# Structure of the complete, membrane-assembled COPII coat reveals a complex interaction network

Joshua Hutchings [1,3], Viktoriya G. Stancheva[2], Nick R. Brown[1,4], Alan C. M. Cheung[1,5], Elizabeth A. Miller [2] & Giulia Zanetti [1✉]

COPII mediates Endoplasmic Reticulum to Golgi trafficking of thousands of cargoes. Five essential proteins assemble into a two-layer architecture, with the inner layer thought to regulate coat assembly and cargo recruitment, and the outer coat forming cages assumed to scaffold membrane curvature. Here we visualise the complete, membrane-assembled COPII coat by cryo-electron tomography and subtomogram averaging, revealing the full network of interactions within and between coat layers. We demonstrate the physiological importance of these interactions using genetic and biochemical approaches. Mutagenesis reveals that the inner coat alone can provide membrane remodelling function, with organisational input from the outer coat. These functional roles for the inner and outer coats significantly move away from the current paradigm, which posits membrane curvature derives primarily from the outer coat. We suggest these interactions collectively contribute to coat organisation and membrane curvature, providing a structural framework to understand regulatory mechanisms of COPII trafficking and secretion.

[1] Institute of Structural and Molecular Biology, Birkbeck College, London, UK. [2] MRC Laboratory of Molecular Biology, Cambridge, UK. [3] Present address: Division of Biological Sciences, University of California San Diego, La Jolla, CA, USA. [4] Present address: The Francis Crick Institute, London, UK. [5] Present address: School of Biochemistry, University of Bristol, Bristol, UK. ✉email: g.zanetti@bbk.ac.uk

Eukaryotic cells are organised in membrane-bound compartments, and a tightly regulated trafficking system ensures proteins and lipids are delivered to the right place at the right time. Cytosolic coat proteins capture secretory cargo and sculpt membrane carriers for intracellular transport[1]. A third of all proteins in eukaryotic cells are synthesised in the Endoplasmic Reticulum (ER) and trafficked to the Golgi, destined for secretion or residency within organelles[2]. ER export is mediated by the coat protein complex II (COPII), which minimally comprises five cytosolic proteins that form two concentric layers on the ER membrane—the inner and outer coat[3]. The inner coat layer consists of the small GTPase Sar1 and the heterodimeric Sec23/24 complex, whilst the outer coat layer comprises the rod-shaped heterotetramer Sec13/31. COPII dynamically assembles and disassembles at ER exit sites, imparting enough force to bend the membrane into unfavourable conformations while at the same time maintaining a range of membrane curvatures. It remains unclear how the COPII coat achieves the required balance of strength and flexibility, and how this balance is regulated to form vesicles of different sizes and shapes, essential to transport a diverse range of cargo molecules[4].

COPII assembly begins with the formation of the inner coat, after GTP-bound Sar1 exposes an amphipathic helix for burial into the ER membrane[5,6]. Sar1-GTP then recruits Sec23/24. Sec23 binds Sar1 and is the dedicated GTPase-activating protein (GAP)[7–9], whilst Sec24 possesses multiple binding sites for cargo recruitment[10,11]. Sec23 also recruits the outer coat subunits Sec13/31 through a flexible proline-rich domain (PRD) in the C-terminal half of Sec31, which accelerates the GAP activity of Sec23[12]. Both the inner and the outer coat are thought to oligomerise and induce membrane curvature to form coated membrane carriers[5,13–15]. COPII has been shown to generate vesicles and tubules both in vivo and in vitro, suggesting the coat is adaptable for different morphologies[8,15–20]. This is consistent with a need to maintain constitutive secretion of soluble proteins, whilst also accommodating much larger cargo such as procollagens and pre-chylomicrons in specialised mammalian cells[4,18,21–23].

COPII assembly is governed by numerous interactions within and between the coat layers[5,12,14,24,25]. Lateral interactions between inner coat subunits mediate its polymerisation into arrays, which have been proposed to prime coat assembly and directly orient membrane curvature through Sar1 amphipathic helix insertion[5]. The outer coat proteins Sec13/31 also self-associate, assembling into cages of different geometries in vitro, including polyhedral and tubular arrangements of varying diameters[13–15]. The assembly units of the cage comprise two structured domains in the N-terminal half of Sec31: an N-terminal β-propeller and an α-solenoid domain. These domains are separated by a blade insertion motif, which binds the Sec13 β-propeller and rigidifies the assembly[26,27]. The Sec31 α-solenoid domain drives homodimerisation of Sec31 to create a rod-shaped tetrameric assembly element, whilst the N-terminal β-propeller domain mediates contacts between four rods to generate a cage[27]. Sec31 also contains a putative helical C-terminal domain (CTD), that is separated from the cage-forming elements by a long flexible PRD[27]. No role for the CTD has yet been assigned, but limited proteolysis experiments and secondary structure prediction suggest an ordered helical domain of ~18 kDa[28,29].

Interactions between inner and outer coat layers are mediated by the Sec31 PRD, which binds Sec23 at several interfaces, including: (i) a GAP-accelerating region that binds the Sar1-Sec23 interface[12], (ii) triple-proline (PPP) motifs binding the tip of the Sec23 gelsolin domain that assist in the assembly of inner coat subunits[5,24], and (iii) a recently defined but structurally uncharacterised charge-based interaction[25]. Several COPII ancillary proteins also possess PRDs that bind Sec23 in a similar way to Sec31, possibly stabilising the coat for formation of larger carriers during procollagen transport in mammals[4,24,30,31]. Some of the interaction interfaces, including outer-inner coat interactions mediated by PPP motifs and the Sec31 active peptide, as well as cage vertices, have been characterised structurally. Disruption of many of these interfaces are tolerated individually but not in combination, implying a network of partially redundant interactions that collectively stabilise coat assembly[5,25]. For instance, partial disruption of outer coat polymerisation by means of an N-terminal His-tag on Sec31 (NHis-Sec31) still permits viability in yeast, but not in combination with other mutations targeting PRD interactions[25].

The full extent and role of coat interactions is not clear, and several questions remain unanswered. How does the interplay between inner and outer coat layers influence membrane curvature and budding morphology? Which coat interactions have a regulatory role? Which interactions are important to provide membrane bending force, and which confer flexibility to the coat? Here, we build on a previously established approach[5,15] using cryo-electron tomography (cryo-ET) and subtomogram averaging (STA) of in vitro reconstituted COPII-coated tubules to obtain the complete, detailed picture of a fully assembled wild-type coat. In addition to structurally characterising known interactions to finer detail, we describe additional ones that link both coat layers into an intricate network.

At the level of the outer coat, we describe a vertex interface that is significantly different from previous reports, we discover an essential role for the structurally and functionally elusive Sec31 CTD as a key node of the COPII network, and an unexpected interaction between Sec31 β-propeller and α-solenoid domains that seems to confer the ability to adapt to membrane with varying curvatures. We map three different interactions between the inner and outer coat layers, including a structurally uncharacterised charged interaction that was recently identified through biochemical and genetic analysis[25]. Finally, at the inner coat assembly interface we resolve a flexible loop on Sec23 that becomes ordered to contribute to lattice formation. We include biochemical and genetic analyses that shed light on the role of many interactions, providing evidence for a complex and flexible network that serves as a basis for dynamic regulation of membrane remodelling.

## Results

### Detailed architecture of outer coat vertices suggests conditional requirement for vertex formation.

Incubating purified COPII components with GUVs and non-hydrolysable GTP (GMP-PNP) induces extensive tubulation of membranes[15,16]. We optimised our previously established in vitro reconstitution and structural analysis pipeline[5,15] to obtain high-resolution cryo-EM data of COPII induced tubules. We collected tilt series of reconstituted budding reactions, which were subsequently used to reconstruct 3D tomograms of the tubules (Fig. 1a and Supplementary Fig. 1a), and we then used STA to obtain the structures, positions and orientations of inner and outer coat subunits (Methods and Supplementary Fig. 1b, d).

The outer coat forms a sparse rhomboidal lattice in which four Sec31 N-terminal β-propeller domains interact to form twofold symmetric X-shaped vertices (Fig. 1b, c, and Supplementary Fig. 3a). The vertex was refined to a resolution of ~12 Å (Supplementary Fig. 2a). We could clearly distinguish the β-propeller shapes and unambiguously rigid-body fit the available Sec13/31 crystal structures (Fig. 1c). The Sec13 β-propeller is also clearly defined, although features gradually degrade along rods further from the vertex, probably due to a higher degree of

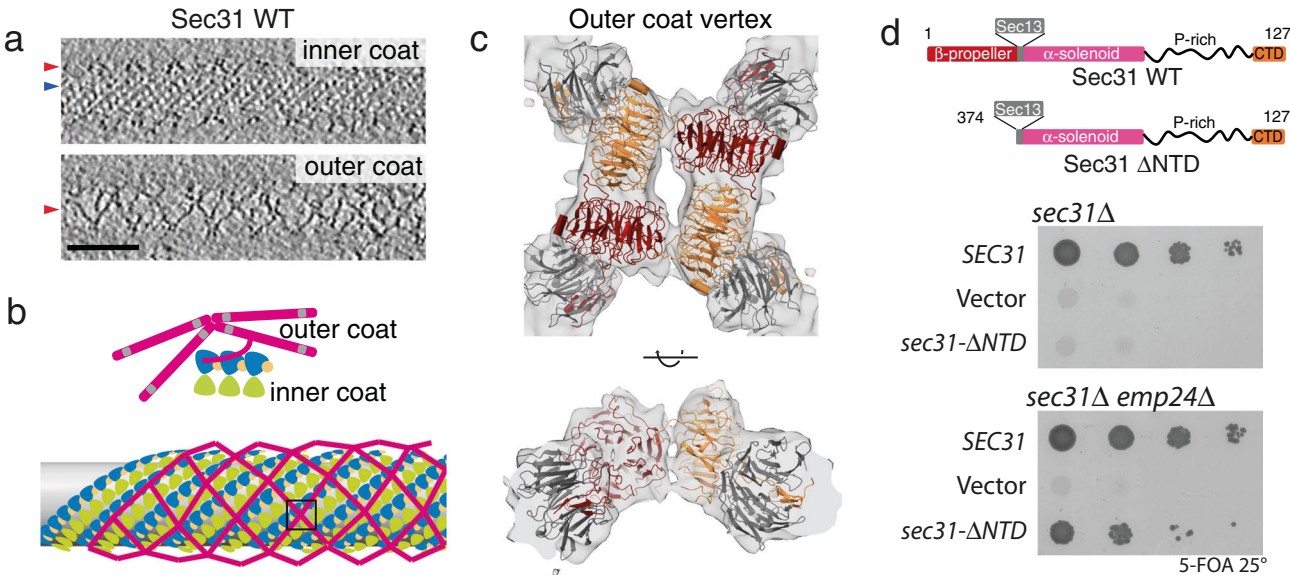

**Fig. 1 Architecture of outer COPII coat vertices. a** Slices through different z heights of a binned and filtered representative cryo-tomogram of wild-type COPII-coated tubule. In the top panel, density for the inner coat lattice is visible from the top and is indicated by a blue arrow, while the outer coat is cut through its side (red arrow). In the bottom panel, the outer coat is visible from the top, with its characteristic lozenge pattern (red arrow). Scale bar 50 nm. These images are representative of tubes seen on 286 tomograms from two separate experiments and data collection sessions. **b** Schematic representation of outer coat architecture. Sec13–31 subunits arrange in a lozenge pattern, around the inner coat that arranges in a tightly packed lattice. **c, d** Focus on the structure of vertices, boxed in this schematics. **c** Top and side views of the 12 Å subtomogram average of the COPII vertex, with rigid-body fitted models (PDB 2PM6 and 2PM9). Sec31 protomers in dark red and orange, Sec13 in grey. **d** Top panel: Domain organisation of wild-type Sec31, and the ΔNTD mutant. Middle panel: a *sec31Δ* yeast strain transformed with the indicated plasmids was tested on 5-FOA, revealing lethality associated with deletion of the NTD. Bottom panel: the same experiment repeated with a *sec31Δ emp24Δ* yeast strain reveals that depletion of emp24 rescues the lethality associated with deletion of the Sec31 NTD. Sigma threshold for contouring was set at 2.2 in (**c, d**). A representative result is shown from at least three replicates.

flexibility. Close analysis of the vertex β-propeller interfaces identified a region of density that likely corresponds to a negatively charged loop (residues 339–357: EQETETKQQE-SETDFWNNV) that is disordered in the crystal structure[27]. It appears that this loop becomes ordered in the assembled vertex and forms an interaction interface with the neighbouring subunits (Supplementary Fig. 3b). We previously discovered that Sec31 with a his-tag at its N-terminus (Nhis-Sec31) yielded tubes with a disordered outer coat, due to destabilization of vertex formation[5]. The proximity of the 339-357 loop to the N-terminus of Sec31 might explain the vertex disruption we observed with Nhis-Sec31[5], as the tag might displace or interfere with this interaction surface.

With this insight into how Nhis-Sec31 might perturb cage assembly, we sought to further probe the importance of vertex interactions by disrupting the system even further and deleting the Sec31 N-terminal β-propeller domain (residues 1-372, Sec31-ΔNTD, Fig. 1d, top panel). Abrogating outer coat vertex interactions completely did not support vesicle formation from microsomes, even with the coat stabilised by non-hydrolysable GTP analogs, a condition that was permissive for Nhis-Sec31[5] (Supplementary Fig. 3c). Sec31-ΔNTD was efficiently recruited to membranes (Supplementary Fig. 3d), and, surprisingly, was capable of tubulating GUVs, suggesting its ability to organise the inner coat array was intact, and that inner coat organisation is sufficient to drive membrane curvature in a synthetic model membrane (Supplementary Fig. 3e). Sec31-ΔNTD was lethal when expressed as the sole copy of Sec31 in wild-type yeast (Fig. 1d, middle panel), but was viable in an *emp24Δ* strain (Fig. 1d, bottom panel). Deletion of Emp24 is thought to lower the membrane bending energy during vesicle formation by depleting abundant lumenally-oriented cargo. This condition has

been shown previously to confer tolerance to the otherwise lethal absence of Sec13[26]. Together, the in vitro and in vivo phenotypes reveal that outer coat vertex interactions are not needed to generate curvature on easily deformable membranes. This suggests that a main driving force for budding is inner coat lattice formation, and that the stable association of vertex interfaces, as well as inner coat stability tuned by GTPase activity, are needed for remodelling of cargo-containing membranes that resist budding.

**Comparison with previously obtained vertex structures.** The arrangement of the Sec31 N-terminal β-propellers in our structure differs significantly from previously published cryo-EM single particle reconstructions obtained from human Sec13–31 cages assembled in the absence of a membrane[13,14,32] (Supplementary Fig. 3f). Indeed, when comparing the soluble cage vertex with that obtained in this study by overlapping one of the Sec31 β-propeller subunits, we find that the relative position of both neighbours is shifted by more than 15 Å (Supplementary Fig. 3g). In the soluble cages, a pair of opposite β-propellers in the vertex forms a tight interaction (identifying the '+' contacts[13]), while the other pair is further apart, separated by the '+' rods (and referred to as the '−' interaction). In the context of the membrane-assembled coat, we see a clear gap between both the '+' and '−' pairs of β-propellers (Supplementary Fig. 3f). Multiple effects might cause this difference: 1. Interactions of vertices arranged in a tubular geometry may be different from those on spherical vesicles; 2. Interactions in soluble cages may be distinct from those in the membrane-assembled coat; and 3. Proteins from different species may have evolved different interaction interfaces, while maintaining an overall similar assembly architecture. We

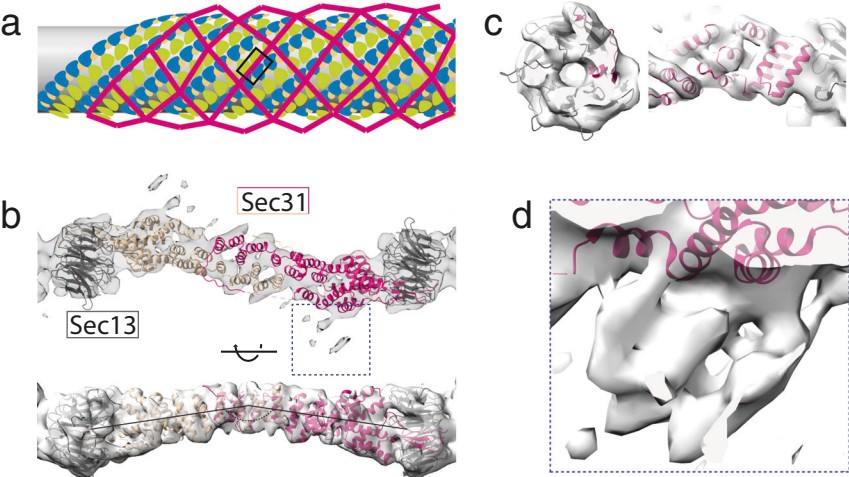

**Fig. 2 Architecture of COPII rods organised in a left-handed manner. a** Schematic representation of the outer coat lozenge architecture, with left-handed rod positions highlighted in the box. **b** Top panel: top view of the 11 Å subtomogram average of the COPII left-handed rod, with rigid-body fitted models (PDB 2PM6). The dimerising protomers are shown in dark and light pink, Sec13 in grey. Bottom panel: side view of (**b**). **c** Details of the map in (**b**), showing the quality of the fit. Left panel: detail of Sec13 β-propeller. Right panel: detail of Sec31 α-solenoid domain. Sigma threshold for contouring was set at 5 for (**b**, **c**). **d** Enlargement of the region boxed in panel 'b' shown at lower contour (sigma = 2.1) to display an extra density.

tested the first two hypotheses by examining the small populations of spherical vesicles and empty cages that were present in our tomograms. We manually picked vertices and performed alignments against two different references: one derived from the soluble cage vertex and one from our vertex structure on membrane tubules. For both datasets, alignments converged to virtually identical structures, with an interface similar to that on membrane-assembled tubules, with a clear gap at the centre of the vertex (Supplementary Fig. 4). While we cannot exclude that the difference we see between empty cages in our sample and those previously published might be caused by buffer conditions, we hypothesise that vertex interactions are different in yeast and human.

**Detailed structure of interconnecting rods**. Outer coat vertices are connected by Sec13–31 rods that wrap tubules both in a left- and right-handed manner, which we refer to as left- and right-handed rods[15]. As mentioned above, when refining the alignment of vertices, the density further from the centre gradually degrades, due to increased flexibility or heterogeneity. We therefore analysed the structure of the interconnecting rods by focusing the refinements at the mid-points between vertices (see Methods). Left-handed rods (Fig. 2a) averaged to a resolution of ~11 Å (Supplementary Fig. 2a), and we could fit the available crystal structures of dimeric edge elements[27] by treating each monomer as a rigid body (Fig. 2b, c). At this resolution we can distinguish helical profiles and individual blades of the Sec13 β-propeller (Fig. 2c). As previously reported, rods in membrane-coated tubules are only slightly bent, resembling the X-ray structure[27] rather than the highly bent edges of soluble assembled cages (Fig. 2b, bottom panel[13]). Surprisingly, we detected a previously unresolved extra density attached to the rod halfway between Sec13 and the dimerisation interface (Fig. 2b, d). The size of this appendage is indicative of a full domain. We reasoned that it could correspond to the Sec31 CTD, which is predicted to be a structured helical domain[27,28]. We could see this extra density clearly only at low contour levels (Fig. 2b, d), indicating either flexibility or sub-stoichiometric binding, which could be a consequence of some domains not being bound, or missing due to degradation (Supplementary Fig. 1e).

**Sec31 C-terminal domain mediates essential coat interactions**. To confirm that the appendage density corresponds to the CTD of Sec31, we analysed GUVs budded with a truncated form of Sec31 (encompassing residues 1 to 1114, referred to as Sec31-ΔCTD, Fig. 3a). In cryo-tomograms of these tubules the outer coat was generally less ordered with respect to the wild type, whilst the inner coat maintained a typical pseudo-helical lattice (Fig. 3b). To conduct an unbiased search for rods, we used a featureless rod-like structure as a template, and subsequently aligned the detected particles to the subtomogram average obtained from the wild-type sample. The average of Sec31-ΔCTD rods recovers the characteristic features and has similar resolution to the wild-type rod, but it lacks the appendage density (Fig. 3c, orange arrowhead), confirming this density most likely corresponds to the CTD. The Sec31-ΔCTD rod also showed weaker density for vertices and the inner coat and membrane layer (Fig. 3c, red, blue and beige arrowheads, respectively), indicative of its less ordered arrangement.

Since no atomic model for the Sec31 CTD has been determined, we built a homology model to fit into the appendage density. Steroid Receptor RNA Activator protein (SRA1)[33] is a functionally unrelated protein that is found only in mammals, and its evolutionary links with Sec31 are unclear. Nevertheless, SRA1 and Sec31 CTD belong to the same evolutionary family and their similarity justifies the use of the SRA1 structure to build a homology model of the Sec31 CTD (see Methods). Rigid-body fitting the homology model in the appendage density shows consistency of size and features, although at this resolution we cannot determine the precise molecular interface (Supplementary Fig. 5a–c).

In order to assess the physiological importance of the Sec31 CTD in the secretory pathway, we made yeast mutants where Sec31 was substituted with Sec31-ΔCTD. When the truncated form was the sole copy of Sec31 in yeast, cells were not viable, indicating that the interaction we detect is essential for COPII coat function (Fig. 3d, left panel). In contrast, when the cargo burden was decreased by deletion of the ER export receptor, Emp24, Sec31-ΔCTD supported viability (Fig. 3d, right panel). This phenotype is similar to the depletion of the Sec31-ΔNTD or of Sec13[26], and leads us to hypothesise that Sec31 CTD binding to Sec31 rods stabilises the COPII interaction network, thereby

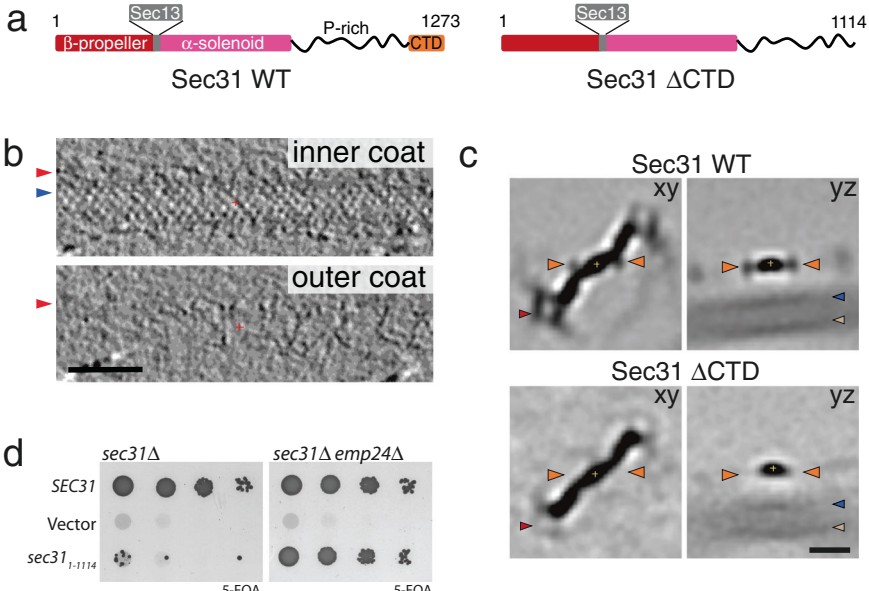

**Fig. 3 Role of the Sec31 CTD. a** Domain organisation of wild-type Sec31, and the ΔCTD mutant. **b** Slices through different z heights of a binned and filtered representative cryo-tomogram of COPII-coated tubule formed with Sec31-ΔCTD. Arrowheads as in Fig. 1a. The outer coat appears partially disordered. Scale bar 50 nm. These images are representative of tubes from 47 tomograms. **c** Top panels: slices through *xy* and *yz* planes of 8× binned subtomogram average of rods extracted from wild-type tubules. Orange arrowheads indicate the appendage density in Fig. 2d. Red arrowhead indicates the vertex, and beige is the membrane. Bottom panels. The same slices for the average of rods extracted from tubules formed with Sec31-ΔCTD. Extra density is absent, confirming its assignment to Sec31 CTD. The density for the vertex and membrane is also fainter, indicating overall disorder of rods in this sample. Scale bar 10 nm. **d** Left panel: a sec31Δ yeast strain transformed with the indicated plasmids was tested on 5-FOA, revealing lethality associated with deletion of the CTD. Right panel: the same plasmids were transformed into a sec31Δ emp24Δ yeast strain revealing rescue of viability. A representative result is shown from at least three replicates.

imparting rigidity and strengthening the coat. Microsome budding reconstitution experiments using Sec31-ΔCTD give further insight into this functional defect. The mutant protein is capable of forming vesicles in the presence of a non-hydrolysable GTP analogue, albeit with reduced efficiency compared to wild type. However, when GTP is used, vesicles fail to form despite Sec31-ΔCTD being efficiently recruited to membranes (Supplementary Fig. 5d, e). This indicates that CTD-mediated outer coat stabilisation becomes necessary when inner coat turnover is allowed, reminiscent of the phenotype seen with Nhis-Sec31 and further supporting a role in outer coat organisation[5].

We next asked whether stabilisation of both the N and C-terminal interactions was dispensable in conditions of efficient inner coat polymerisation, by performing GUV budding reconstitutions in the presence of Nhis-Sec31-ΔCTD. While both Sec31-ΔCTD and Nhis-Sec31 showed tubulation, we could not detect any tubules in negatively stained grids of the combined mutant. This suggests that, even when membranes are easily deformable and inner coat assembly is stabilised by the absence of GTP hydrolysis, some level of outer coat organisation is required to deform membranes, and the inner coat bridging activity of the Sec31 triple-proline motifs is not sufficient.

**Interactions between β-propeller and α-solenoid domains define extra rod connections**. We next analysed rods that interconnect vertices in the right-handed direction (Fig. 4a). Surprisingly, in addition to the CTD appendages, a second region of ill-defined extra density was present near the Sec31 dimerisation region (Supplementary Fig. 6a). Upon classification we could divide the right-handed rod dataset into two classes, both of which converged to resolutions between 13 and 15 Å (Fig. 4b and Supplementary Fig. 3b). The first class is analogous to the left-handed rods, whereas the second class has a density attached to

the centre of the rod which clearly resembles a pair of β-propeller subunits, suggesting the presence of an unexpected additional Sec13–31 edge attached to the Sec31 α-solenoid. We confirmed the nature of the extra density by focussing refinements on the predicted centre of the 'extra' rod, obtaining the unambiguous shape of a Sec13–31 heterotetramer (Supplementary Fig. 6b). This runs nearly perpendicular to, and bridges between, two right-handed rods. Analysis of the averages placed in the context of individual tomograms shows that the extra rods are sparsely and randomly distributed (Fig. 4c). We also note that they follow a similar direction to the left-handed rods, running along the direction of the main Sec23-Sec23 inner coat interfaces (Fig. 4d).

Since the extra rods bridge between α-solenoid dimerisation interfaces of right-handed rods, we expect that the distribution of neighbouring vertices compared to the centres of the extra rods should form a rhombus (dotted lines in Supplementary Fig. 6c). However, when we plotted the position of vertices neighbouring the extra rods, we noticed that in addition to the expected peaks, there was a cluster of vertices positioned at the tip of the extra rods (Supplementary Fig. 6c, red circle). This suggested that a subpopulation of these extra rods could connect to a right-handed rod on one side and form a standard vertex on the other. By selecting these rods and calculating their average, we confirmed the presence of the two different connections (Supplementary Fig. 6d). The localisation of these hybrid rods in the tomogram shows that these are often placed at the interface between two patches of outer coat lattice that come together with a mismatch (Supplementary Fig. 6e). This indicates that the mode of interaction we characterise here might help the outer coat network adapt to different curvatures.

**Extended interactions between the inner coat and the Sec31 disordered region**. Compared to previous work[5], the ordered

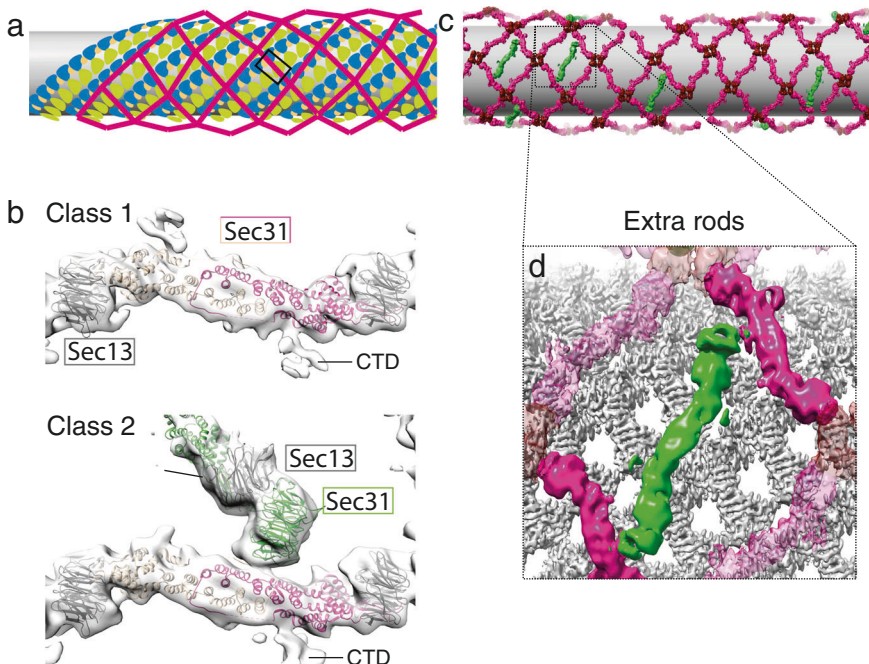

**Fig. 4 Architecture of COPII rods organised in a right-handed manner. a** Schematic representation of coat architecture, with right-handed rod positions highlighted in the box. **b** Top panel: top view of the 13 Å subtomogram average of the COPII right-handed rod with no extra density at the dimerisation interface, with rigid-body fitted models (PDB 2PM9). The dimerising protomers are shown in dark and light pink, Sec13 in grey. The CTD is visible in these rods. Bottom panel: the same view of a class of right-handed rods that displays an attachment of another rod. The Sec13–31 beta propeller tandem is fitted (Sec31 in green and Sec13 in grey). Sigma threshold for contouring was set at 2.6. **c** Averages of vertices (dark red), left and right-handed rods (pink) are placed in their aligned positions and orientations in a representative tomogram. Extra rods are placed in green, and display random distribution, but unique orientation. **d** Enlargement of the placed object, with the placed inner coat (grey), showing that extra rods orient roughly parallel to the inner coat lattice direction.

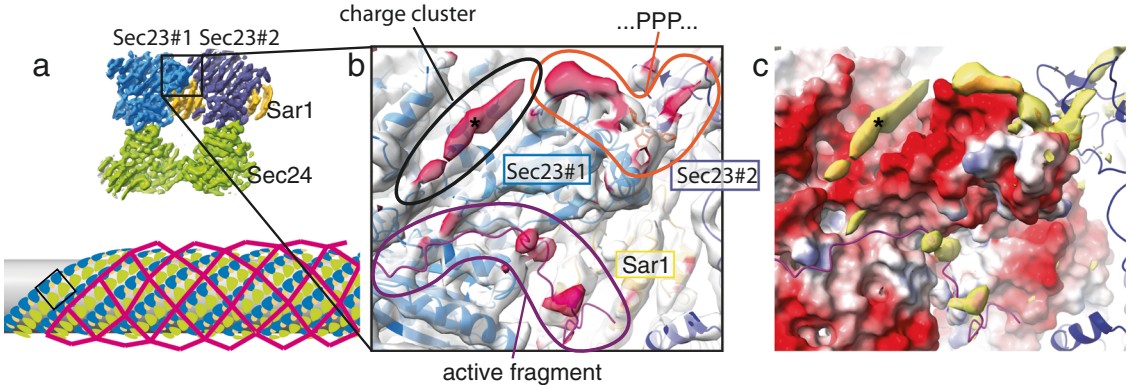

**Fig. 5 Inner-outer coat interactions. a** Top: the inner coat lattice structure, with a box indicating the region represented in (**b**, **c**). Bottom: schematic representation of coat architecture, with inner coat highlighted in the box. **b** The density-modified inner coat subtomogram average and the refined model. The map is coloured in transparent white, while regions that are further than 3 Å from the model are coloured in dark red, indicating density that is not explained by the model and is attributed to outer coat binding. Sigma threshold for contouring was set at 0.08. **c** The difference map between the map and a model-generated density (filtered at 4 Å, yellow) is overlaid to the model surface coloured according to its Coulombic potential. The outer coat density indicated by the asterisk binds to a negatively charged groove on Sec23.

outer coat now allows us to gain insights into inter-layer interactions. We therefore refined the structure of the inner coat from fully ordered coated tubules to an average resolution of 4.6 Å (Supplementary Fig. 2b). Density modification[34] and sharpening further improved features in the map (Supplementary Fig. 7a), permitting to unambiguously fit X-ray models of Sec23, Sec24 and Sar1. The resolution and overall quality of our map allowed us to build regions that were missing from the X-ray structures and refine the model (Supplementary Fig. 7b, c). We analysed the sites of interaction with the outer coat by identifying regions in

the map that are not explained by the model (prominent regions in their difference map, Fig. 5a, b). These are generally better defined than in our previous map obtained with disordered outer coat[5], possibly due to more stable interactions and lower flexibility. We confirmed the binding of the Sec31 active peptide to Sec23 through its WN residues, as well as that of Sec31 PPP motifs to the Sec23 gelsolin domain. Here we can clearly detect a single density that extends on both sides of the prolines to contact two adjacent inner coat subunits, supporting previous hypotheses that PPP-containing sequences bridge between neighbouring

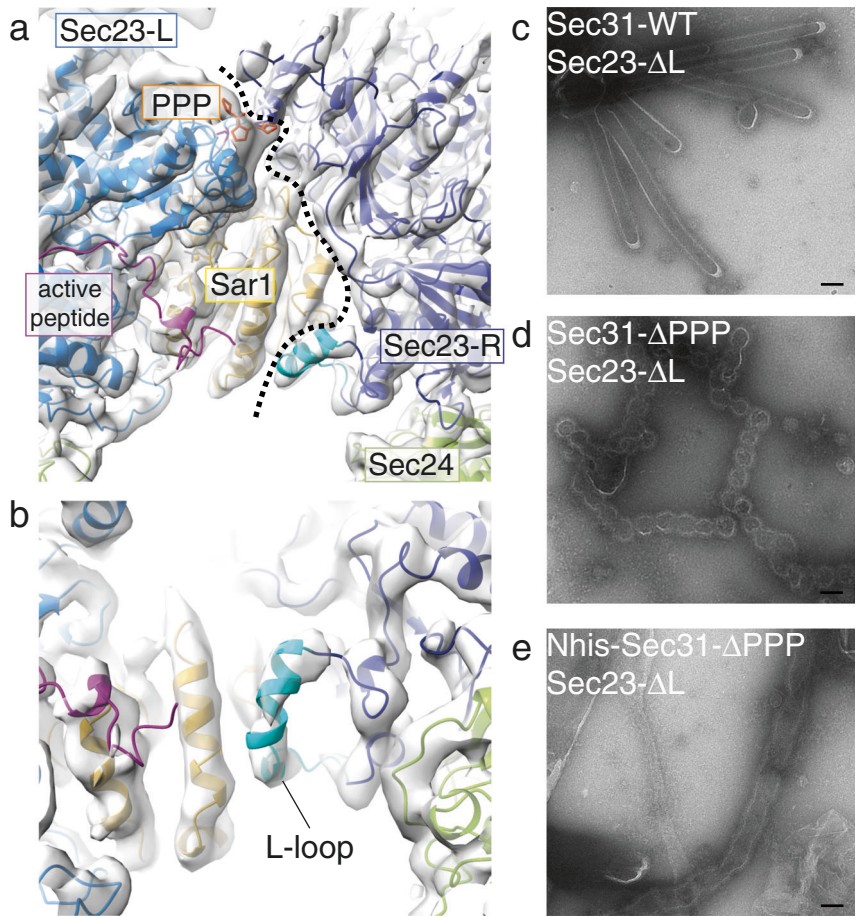

**Fig. 6 Interface between inner coat subunits. a** The interface between neighbouring inner coat subunits viewed from the top. A dotted line defines the boundary between two protomers. **b** Enlarged view of the Sec23-Sar1 interface, with the modelled Sec23 17-residue L-loop in dark cyan. This loop is disordered in the X-ray structure, but becomes ordered in the lattice, mediating interactions with Sar1. Sigma threshold for contouring was set at 0.08 for (**a**, **b**). **c** GUV budding reconstitutions in the presence of wild-type Sec31 and Sec23 lacking the L-loop show straight tubules, similar to wild-type reactions. **d** Budding creates spherical profiles when Sec31 has all of its PPP sites mutated to alanine to further weaken inner coat assembly. **e** Floppy tubules form when, in addition to the conditions in 'd', Sec31 vertex interactions are destabilised with Nhis-Sec31. Scale bar 100 nm. Micrographs in (**c-e**) are representative of experiments repeated at least three times.

inner coat subunits and contribute to the stability of the lattice (Fig. 5b)[5,24,25].

In addition to the expected Sec31 binding sites, the difference map showed a prominent region that we did not see in the context of Nhis-Sec31. We now see density corresponding to a long 'sausage-like' region nestled in a negatively charged concave surface between the Sec23 Zn-finger, helical, and gelsolin-like domains (Fig. 5b, c, asterisk). In a recent report, we showed that binding between the outer and inner layers of the COPII coat is mediated by multivalent interactions of the Sec31 disordered domain with Sec23[25]. These interactions involve the previously identified catalytic and triple-proline regions, and a charge-based interaction between the positively charged Sec31 disordered domain and negatively charged surface on Sec23. Charge reversal of the Sec23 surface led to abolished recruitment of Sec13–31 and a non-functional coat[25]. We are now able to map this essential interaction and show it spans ~25 Å, corresponding to 9–10 residues.

**Sec23–Sar1 interactions in lattice formation**. Inner coat subunits assemble through a lattice interface between Sar1 and Sec23 from one protomer, and Sec23 from the neighbouring protomer (Fig. 6a). Analysis with the PDBePISA web server[35], indicates this interface extends over a large surface of 910 Å², with individual

contacts expected to only partially contribute to its stability. The PPP-mediated contacts between one Sec23 gelsolin-like domain and the neighbour Sec23 Zn-finger domain are part of this extended interface. As part of the extended lattice interface, Sar1 contacts the neighbouring Sec23 trunk domain. We detect a prominent contribution mediated by a 17-residue loop of Sec23 (residues 201–217, KPTGPGGAASHLPNAMN, which we name L-loop, for lattice) that is not visible in the X-ray structures. Secondary structure predictions denote this region as disordered and highly prone to protein binding (Supplementary Fig. 7d). In our structure we can clearly visualise and model the L-loop in Sec23, which becomes partially ordered in its interaction with Sar1 (Fig. 6b). To assess the importance of the L-loop interaction in lattice formation and membrane tubulation, we mutated the 17 residues to a stretch of 5 glycine-serine repeats and tested this mutant in GUV budding reactions. The Sec23 L-loop mutant did not lead to any significant phenotype, with straight tubes forming (Fig. 6c). This is not unexpected, due to the loop's marginal contribution to the inner coat lattice interface. Indeed, weakening the lattice interface by mutating PPP motifs on Sec31 (Sec31-ΔPPP) does not change tube morphology either[25]. However, when we aggravated the disruption by combining the Sec23 L-loop and Sec31-ΔPPP mutants, budding reactions showed a striking absence of straight tubules, and enrichment of

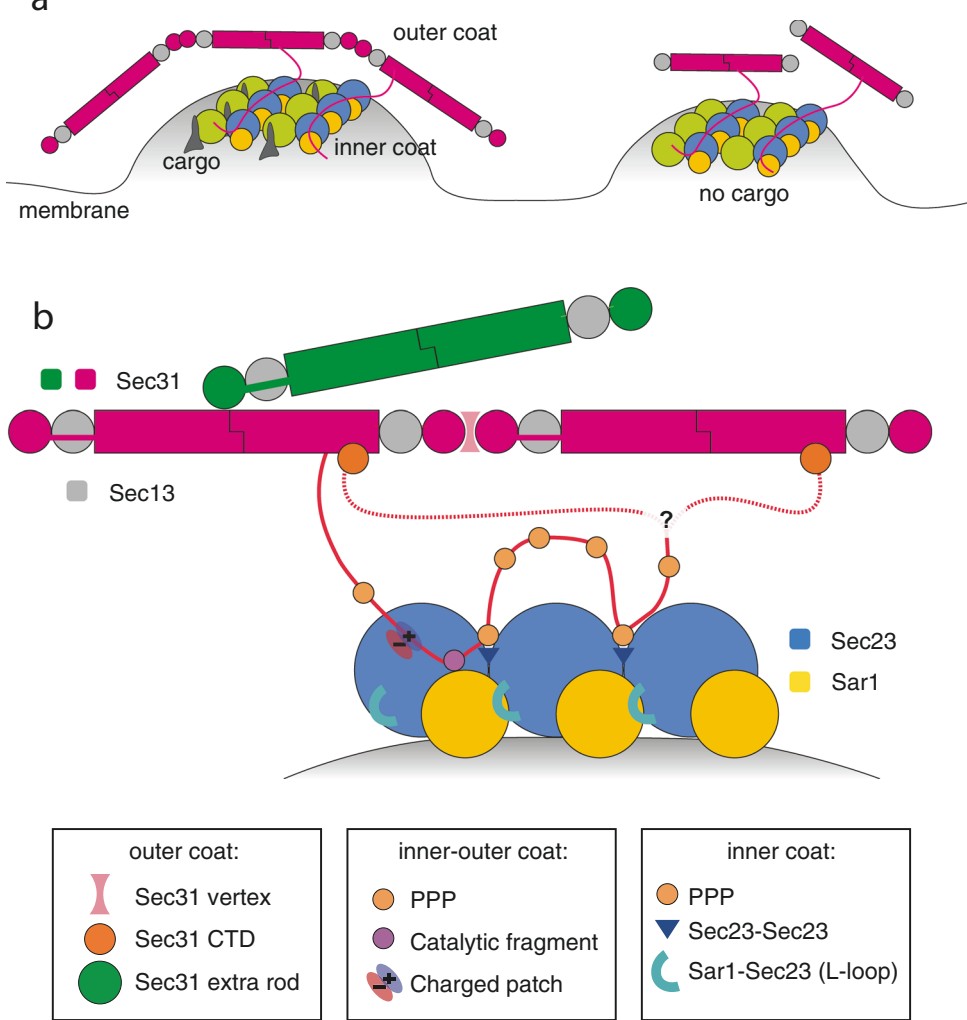

**Fig. 7 A map of the COPII coat assembly network.** A schematic model for how COPII assembles on membranes. **a** Overview of COPII organisation on budding ER membrane. Left panel: a fully functional COPII coat promotes budding of cargo-containing membranes. Right panel: a coat whose outer coat is unable to polymerise to form cages is still able to induce budding of cargo-replete, easily deformable membranes by promoting inner coat assembly. **b** Details of three sets of interactions contribute to coat assembly: 1. Outer-outer coat. Mediated by Sec31 β-propellers forming vertices, by Sec31 β-propellers binding to the α-solenoid domain of a different protomer to create 'bridging' rods, and by Sec31 CTD binding to Sec31 α-solenoid domain. It is unclear whether the latter interaction occurs in cis or trans. 2. Outer-inner coat. Mediated by Sec31 disordered region, contributing three interaction sites: triple-proline motifs bind to Sec23 bridging between neighbouring subunits; the active peptide binds across Sec23 and Sar1 to accelerate GAP activity, and positively charged clusters bind to a negatively charged groove on Sec23. 3. Inner–inner coat. Mediated by Sec31 PPP motifs (see above), and by an extended lattice interface which includes Sec23–Sec23 interactions as well as Sec23–Sar1 interactions mediated by the L-loop.

multibudded profiles (Fig. 6d). This indicated that when inner coat interactions are significantly weakened, the outer coat becomes the main determinant of membrane remodelling and defaults to inducing spherical curvature. Indeed, when we also weakened outer coat vertex interactions by using an N-terminal his-tagged version of Sec31-ΔPPP, budding gave rise to 'floppy' tubules rather than multibudded profiles (Fig. 6e).

## Discussion

We have combined cryo-tomography with biochemical and genetic assays to obtain a complete picture of the assembled COPII coat in fine detail. We make a number of observations which allow us to piece together a picture of the COPII coat as a complex network of partially redundant interactions (Fig. 7). Structural and functional analysis of each interface reveals their role in coat assembly and membrane remodelling, shedding light on the COPII-mediated membrane budding mechanism.

We map outer coat vertex interactions in detail, and dissect their role in membrane remodelling. Using an N-terminal deletion mutant of Sec31 in a range of assays that sample various degrees of membrane deformability, we show that outer coat assembly into cages is necessary in vivo to overcome membrane resistance to deformation, but is dispensable in conditions where membranes are more easily deformed thanks to the absence of certain cargo (Figs. 1, 7a and Supplementary Fig. 3). Together with the previous report that outer coat cage stability is dispensable in vitro when inner coat turnover is inhibited[5], our results challenge the widely accepted role of the outer coat as a main driver of membrane curvature.

The interactions we observe between four Sec31 β-propeller subunits at the vertices of the outer coat lattice (Supplementary Fig. 3) are distinct from the analogous vertices seen in human Sec31 cages assembled in vitro in the absence of a membrane and Sar1[13,14]. The vertex structure we report here for yeast proteins is much less compact, with deviations of over 15 Å in the relative

positions of neighbouring β-propellers. We saw this arrangement on spherical vesicles as well as empty cages, leading us to hypothesise that the vertex is more compact in humans than in yeast. It will be interesting to assess whether organisms with more complex secretory needs have selected tighter and more stable interactions at vertices.

Thorough structural analysis of the Sec13/31 rods reveals previously uncharacterised interactions between outer coat units. The elusive CTD of Sec31 forms a helical bundle, and its function was unknown to date. We show that Sec31 CTD is as an essential node of the outer coat network that binds to Sec31 α-solenoid domains (Figs. 2, 3 and Supplementary Fig. 5). While we cannot assign a definite function to the Sec31 CTD, the fact that it is dispensable when membranes are made more deformable by depletion of certain classes of cargo is reminiscent of the role of Sec13[26], and of Sec31-ΔNTD, and suggests that Sec31 CTD contributes to coat rigidity and/or stability. One possibility is that CTD binding has a role in restricting the outer coat freedom to move once bound to the inner coat through its flexible disordered domain, thereby increasing the probability that outer coat lattice can form. Consistent with this, the outer coat on tubules reconstituted with Sec31-ΔCTD appears less ordered than in wild-type conditions. Interestingly, human Sec31 proteins lacking the CTD assemble into cages[29], indicating that either the vertex is more stable for human proteins, or that the CTD is important in the context of membrane budding but not for cage formation in high salt conditions. Our data paints a picture of the assembled coat where the outer coat C-terminal disordered region reaches down to bind and stabilise the inner coat, and then loops back to lock onto the outer coat lattice. Our data does not distinguish between a scenario in which C-terminal domains interact in *cis* or *trans* with outer coat rods, but it is interesting to hypothesise that trans-interactions might further stabilise the coat network (Fig. 7). While disruption of outer coat assembly at either Sec31 N-terminus or C-terminus is conducive to budding in conditions of high membrane deformability (for example budding GUVs), when both interactions are disrupted by using a Nhis-Sec31-ΔCTD construct budding is inhibited, indicating that some level of outer coat assembly is required for membrane deformation.

We also discover a second interaction within the outer coat: in addition to the known interaction of Sec31 β-propellers with each other at vertices, these domains can also bind to the Sec31 dimerisation interface, at the centre of the α-solenoid region. This leads to extra outer coat rods creating a bridge between canonical rods (Fig. 4). Occasionally these extra rods form a canonical vertex interaction at one end, and an orthogonal interaction with other rods at the other end. Such rods 'glue' mismatched patches of outer coat lattice together: they might therefore be important for outer coat stabilisation in a context of flexibility and adaptability (Supplementary Fig. 6). It is interesting that we could only detect extra rods running in the left-handed direction, and connecting canonical right-handed rods. This could be explained by the scenario in which the Sec31 disordered PRD binds to multiple Sec23 subunits in tandem, leading to preferential orientation of the extra rods with respect to the inner coat. Due to limited particle numbers, the resolution we obtained does not allow us to precisely define the residues involved in the interaction between the β-propeller and the α-solenoid domains of Sec31. Higher resolution will be needed to inform mutational analysis and assess the physiological and functional relevance of this connection.

Interactions between the outer and inner coat are mediated by Sec31 disordered PRD[12,24,25,27]. By analysing the structure of the inner coat we confirm two interactions that have been previously defined and characterised structurally (Fig. 5): firstly, the Sec31 'active peptide' binds across Sec23 and Sar1, contributing residues in proximity to the GTP binding pocket, and accelerating Sec23

GAP activity[12]. Secondly, Sec31 contains triple-proline motifs, shared in metazoa by other COPII-interacting factors such as TANGO1 and cTAGE5[24]. These residues bind to the Sec23 gelsolin-like domain and appear to bridge adjacent inner coat subunits, aiding inner coat lattice formation[5,24]. In addition, an essential interaction between outer and inner coat layers was recently discovered. This is mediated by the negatively charged surface of Sec23 that was postulated to interact with positively charged clusters within the Sec31 PRD[25]. We detect a prominent density bound to this region of Sec23, located within a groove formed at the junction between the gelsolin, helical and Zn-finger domains. We attribute this density to the interacting Sec31 positively charged regions (Fig. 5). Features of this extra density are less well-defined compared to the rest of the protein. Since multiple charge clusters in yeast Sec31 may contribute to this interaction interface[25], the low resolution could be explained by the fact that the density is an average of different sequences. Although we previously observed densities corresponding to the PPP and active peptide interactions in our structure assembled with Nhis-Sec31, the charged interaction was not detected[5]. It is possible that ordering of the outer coat into a lattice improves the stability and occupancy of this interface.

Finally, interactions mediating inner coat assembly into a lattice are defined by our analysis. The first is the extended interface between Sec23 protomers that involves interaction with a neighbouring Sec23/Sar1 dimer and is bridged by Sec31 PPP motif (Fig. 6a). The second interface is a previously unknown interaction between Sec23 and a neighbouring Sar1 molecule, mediated by a 17-residue loop (L-loop) in Sec23 which becomes ordered upon formation of the inner coat lattice (Fig. 6b), and whose importance in inner coat lattice assembly was confirmed biochemically.

Disrupting the inner coat lattice interface in combination with a Sec31 competent for outer coat assembly leads to budding of vesicles with spherical profiles, rather than a majority of straight tubules (Fig. 6d), indicating that outer coat assembly into cages dictates spherical membrane shape when the inner coat is unstable. This might be reflected in a physiological scenario where GTP-hydrolysis triggers inner coat turnover by removing Sar1 from the membrane, favouring spherical vesicles. Metazoan proteins such as TANGO1 and cTAGE5, which contain PPP motifs but do not accelerate GTP hydrolysis, could work by stabilising the inner coat interface while inhibiting GTP hydrolysis, favouring tubules and promoting transport of large carriers such as procollagen[24]. Sec23 is a highly conserved protein, and is present in two paralogues in metazoa: Sec23A and B. While the two paralogues are thought to have largely redundant functions, mutations in Sec23A but not B cause defects in secretion of procollagen[17,36]. Human Sec23A and B are 85% identical, and the L-loop sequence is a region that varies significantly (Supplementary Fig. 7e). Because this region is important in stabilising lattice formation, we hypothesise it is involved in promoting formation of large carriers: the difference between Sec23A and B in the L-loop might confer an ability to differentially support large carrier budding, and explain their distinct roles in procollagen export disease.

We know from previous studies that partial disruption of both inner and outer coat layers is incompatible with life, but can be rescued by relieving the cell of bulky cargoes[25,26]. Here we show that weakened coat interactions at the level of both inner and outer coat leads to the formation of floppy tubules (Fig. 6e), suggesting the coat does partially assemble and impart some membrane deformation, but not sufficient for active cargo transport in cells. Together, these data suggest that a balance of lattice contacts between the inner and outer coats support membrane budding, and this balance can be tuned to achieve different morphologies depending on membrane deformability.

In summary, we have shown that COPII forms a complex network that assembles through partially redundant interactions, whose effects are combined for a productive budding event. We have obtained a detailed map of this network and have shown that the extent to which the presence and stability of each interaction are necessary depends on the membrane deformability. This makes the COPII system an ideal platform for regulation in response to dynamically changing cargo requirements, such as its abundance, shape and size.

## Methods

**Cloning**. Yeast COPII components were cloned from the *Saccharomyces cerevisiae* S288c genome into appropriate expression vectors using In-Fusion (Takara), specifically: pETM-11 (AddGene) for Sar1 and pFASTBacHTb (AddGene) for Sec23/24 and Sec13/31. N-terminal hexa-histidine purification tags were cloned with an intervening TEV protease cleavage site for Sar1, Sec24 and Sec31. Primers are listed in Supplementary Table 1.

**Protein expression and purification**. Sar1: The pETM-11-Sar1 construct was transformed into BL21 using standard heat shock methods. Two litres of BL21 were induced with 1 mM IPTG for 3 h at 25 °C before harvesting. Sar1 was affinity purified following application to a 5 mL HisTrap column (GE Healthcare) equilibrated in lysis/binding buffer (50 mM Tris-HCl, 150 mM NaCl, 0.1% Tween-20 (v/v), 10 mM imidazole, 1 mM DTT, pH 8.0). Elution was achieved with a linear gradient of elution buffer (as for binding buffer, with 500 mM imidazole). Pure fractions were pooled and incubated with TEV protease at a 1:50 ratio of protease: Sar1 (w/w) in a sealed 10 kDa MWCO dialysis tube submerged in two litres of HisTrap binding buffer for overnight dialysis at 4 °C. The dialysed product was reapplied to the HisTrap column with the flow-through collected and concentrated to ~0.7 mg/mL, determined using a Bradford assay.

Sec23/24: One litre of Sf9 insect cells (at $1 \times 10^6$ cells/mL) were infected with baculovirus: 9 mL/L of untagged Sec23p and 3 mL/L of His-tagged Sec24p. These were incubated for 3 days at 27 °C and 100 rpm shaking. Cells were harvested using a glass homogeniser and centrifuged at $167{,}424 \times g$ for 1 h at 4 °C. Sec23/24 was affinity purified following application to a 5 mL HisTrap column (GE Healthcare) equilibrated in lysis/binding buffer (20 mM HEPES (pH 8.0), 250 mM sorbitol, 500 mM potassium acetate, 10 mM imidazole, 10% glycerol and 1 mM DTT). Elution was achieved with a linear gradient of elution buffer (as for binding buffer, with 500 mM imidazole). Pure fractions were collected and diluted approximately twofold with low salt anion-exchange binding buffer (20 mM Tris, 1 mM magnesium acetate, 0.1 mM EGTA, and 1 mM DTT, pH 7.5) before application to an equilibrated 5 mL HiTrap Q column (GE Healthcare). Elution was achieved with a linear gradient of elution buffer (as for binding buffer, with 1 M NaCl). Pure fractions were pooled and diluted to ~1.26 mg/mL with low salt buffer and 10% glycerol, which were then aliquoted, flash-frozen and stored at −80 °C. The same protocol was applied for the Sec23-ΔL loop mutant. For Sec23-ΔL, residues 201–218 were mutated to 5xGS repeats using Sec23 pFastBacHTb as the template. The mutation was amplified by PCR and incorporated using In-Fusion. The same protein expression and purification protocol as WT Sec23/24 was used.

Sec13/31: One litre of Sf9 insect cells (at $1 \times 10^6$ cells/mL) were infected with baculovirus: 9 mL/L of untagged Sec13p and 3 mL/L of His-tagged Sec31p. To obtain His-tagged Sec13/31 (Sec13/31-NHis), the same cell lysis and purification protocol (including buffers) was followed as for Sec23/24. Pure fractions from anion exchange were pooled and concentrated to ~2 mg/mL for 50 µL aliquots, which were flash-frozen and stored at −80 °C.

For cleaved Sec13/31p an additional overnight TEV protease cleavage step was performed for His-tag removal prior to anion exchange. Following the initial affinity purification with a 5 mL HisTrap column (GE Healthcare), the pooled eluate was incubated with TEV protease at a 1:50 ratio of protease:Sec13/31 (w/w) in a sealed 10 kDa MWCO dialysis tube submerged in two litres of HisTrap binding buffer for overnight dialysis at 4 °C. The dialysed product was reapplied to the HisTrap column, the flow-through was collected, and then diluted approximately fourfold with low salt buffer for application to a 5 mL HiTrap Q column. Elution was achieved with a linear gradient of elution buffer (same as Sec23/24 Q elution buffer). Pure fractions were pooled and concentrated to ~2 mg/mL for 50 µL aliquots, which were flash-frozen and stored at −80 °C.

On the day of GUV BR preparation, Sec13/31 and/or NHis-Sec13/31 aliquots were thawed and gel-filtrated on a 2.4 mL Superdex200 column (GE Healthcare) mounted on a ÄktaMicro (GE Healthcare) system, equilibrated in HKM buffer (20 mM HEPES, 50 mM KOAc and 1.2 mM MgCl2, pH 6.8). Cleaved and uncleaved versions of the Sec13/31 mutants used in this study were prepared in the same as the wild-type protocol detailed above.

For the Sec13/31-ΔNTD, a C-terminally His-tagged version of Sec31 truncated at position 373 was used. This was derived from the Sec31-NHis pFASTBacHTb construct using In-Fusion cloning. For the Sec13/31-ΔCTD, a N-terminally His-tagged version of Sec31 covering residues 1-1114 was used. This was derived from the Sec31-NHis pFASTBacHTb construct using In-Fusion cloning. For Sec13/31-ΔPPP, a synthetic DNA construct with seven PPP to SGS mutants (853–855,

965–967, 966–968, 981–983, 1042–1044, 1096–1098 and 1107–1109) was inserted into the Sec31-NHis pFASTBacHTb construct using In-Fusion.

**GUV budding reactions**. Giant unilamellar vesicles (GUVs) were prepared by electroformation[37] from 10 mg/mL of the "major–minor" lipid mixture[38] suspended in a 2:1 chloroform:methanol solvent mix, as described previously[5]. The mixture is spread over two Indium Tin Oxide (ITO)-coated glass slides, which are sandwiched with a silicon spacer to create a chamber that is then filled with 300 mM sucrose. An alternating voltage of 10 Hz and 3 V (rms) was applied for 6–8 h using copper tape attached to the ITO-coated slides. GUVs were harvested by gentle aspiration from the chamber and applied to 500 µL of 300 mM glucose for gravity sedimentation overnight at 4 °C. The supernatant was carefully aspirated and discarded to leave a ~30–50 µL GUV pellet the next day. GUVs were used within 2 days of harvesting.

For GUV budding reactions (BRs), COPII proteins were incubated at defined concentrations (1 µM Sar1p, 320 nM Sec23/24p, 173 nM Sec13/31p) with 1 mM GMP-PMP (Sigma-Aldrich), 2.5 mM EDTA (pH 8.0) and 10% GUVs (v/v). The same concentrations were used for BRs with mutant COPII components. Reactions were left at room temperature for 1–3 h prior to negative stain or vitrification.

**Yeast strains and plasmids**. Yeast strains and plasmids used in this study were all generated using standard molecular biology techniques. The yeast strains used were previously published: LMY1249 (*sec31::NAT pep4::TRP ade2-1 his3-11 leu2-3,112 + [pYCp50::SEC31-URA3]*) described[5] and VSY015 (*sec31::NAT emp24::KANMX pep4::TRP ade2-1 his3-11 leu2-3,112 + [pYCp50::SEC31-URA3]*) described[25]. Plasmids were introduced into yeast using standard LiAc transformation methods. Cultures were grown at 30 °C in standard rich medium (YPD: 1% yeast extract, 2% peptone, and 2% glucose) or synthetic complete medium (SC: 0.67% yeast nitrogen base and 2% glucose supplemented with amino acids) as required. For testing viability, strains were grown to saturation in SC medium selecting for the mutant plasmid overnight at 30 °C. Tenfold serial dilutions were made in 96 well trays before spotting onto 5-FOA plates (1.675% yeast nitrogen base, 0.08% CSM, 2% glucose, 2% agar, 0.1% 5-fluoroorotic acid). Plates were scanned at day 2 or day 3 after spotting and growth at 30 °C. The *SEC31* WT plasmid used in this study (VSB49) was described[25] and consists of *SEC31* PCR-amplified with native 500 bp upstream and downstream of the gene and cloned in BamHI/NotI sites of pRS313 (*HIS, CEN*[39]). Truncations were introduced via site-directed mutagenesis using the QuikChange system (Agilent) and Gibson Assembly (New England Biolabs) as per manufacturers' instructions to generate VSB146 (stop codon at position 1114) and VSB131 (deletion of Val2-Gln372). All experiments were repeated three times and a representative is shown.

**Liposome binding**. Liposome binding experiments were performed as described[40]. Briefly, synthetic liposomes of 'major/minor' composition (50 mol% phosphatidylcholine, 21 mol% phosphatidylethanolamine, 8 mol% phosphatidylserine, 5 mol % phosphatidic acid, 9 mol% phosphatidylinositol, 2.2 mol% phosphatidylinositol-4-phosphate, 0.8% mol% phosphatidylinositol-4,5-bisphosphate, 2 mol% cytidine-diphosphate-diacylglycerol, supplemented with 2 mol% TexasRed-phosphatidylethanolamine and 20% (w/w) ergosterol) were dried to a lipid film in a rotary evaporator and rehydrated in HKM buffer (20 mM HEPES pH 7.0, 160 mM KOAc, 1 mM MgCl2). The lipid suspension was extruded 17 times through a polycarbonate filter of 0.4 µm pore size. Purified COPII components and lipids were mixed to final concentrations of 0.27 mM liposomes, 15 µg/ml Sar1, 20 µg/ml Sec23/Sec24, 30 µg/ml Sec13/Sec31 and 0.1 mM nucleotide in 75 µl HKM Buffer. Binding reactions were incubated for 30 min at 25 °C. Each sample was mixed with 50 µl 2.5 M Sucrose-HKM, then 100 µL transferred to an ultracentrifuge tube, overlaid with 100 µl 0.75 M Sucrose-HKM and 20 µl HKM. The gradients were spun ($434{,}513 \times g$, 25 min, 24 °C with slow acceleration/deceleration) in a Beckman TLA-100 rotor. The top 30 µl of the gradients were collected and normalised for lipid recovery using Typhoon FLA 7000 scanner (GE). Samples were then resolved by SDS-PAGE and visualised using SYPRO Red staining. All experiments were repeated three times and a representative is shown.

**Microsome budding assays**. Microsomal membranes were prepared from yeast as described[41]. Briefly, yeast cells were grown to mid-log phase in YPD (1% yeast extract, 2% peptone, and 2% glucose), harvested and resuspended in 100 mM Tris pH 9.4/10 mM DTT to 40 OD$_{600}$/ml, then incubated at room temperature for 10 min. Cells were collected by centrifugation and resuspended to 40 OD$_{600}$/ml in lyticase buffer (0.7 M sorbitol, 0.75X YPD, 10 mM Tris pH 7.4, 1 mM DTT + lyticase 2 µL/OD$_{600}$), then incubated at 30 °C for 30 min with gentle agitation. Cells were collected by centrifugation, washed once with 2X JR buffer (0.4 M sorbitol, 100 mM KOAc, 4 mM EDTA, 40 mM HEPES pH 7.4) at 100 OD$_{600}$/ml, then resuspended in 2X JR buffer at 400 OD$_{600}$/ml prior to freezing at −80 °C. Spheroplasts were thawed on ice, and an equal volume of ice cold dH20 added prior to disruption with a motor-driven Potter Elvehjem homogeniser at 4 °C. The homogenate was cleared by low-speed centrifugation and crude membranes collected by centrifugation of the low-speed supernatant at $27{,}000 \times g$. The membrane pellet was resuspended in ~6 mL of buffer B88 (20 mM HEPES pH 6.8, 250 mM sorbitol, 150 mM KOAc, 5 mM Mg(OAc)$_2$) and loaded onto a step sucrose gradient

composed of 1 mL 1.5 M sucrose in B88 and 1 mL 1.2 M sucrose in B88. Gradients were subjected to ultracentrifugation at 190,000 × g for 1 h at 4 °C. Microsomal membranes were collected from the 1.2M/1.5M sucrose interface, diluted tenfold in B88 and collected by centrifugation at 27,000 × g. The microsomal pellet was resuspended in a small volume of B88 and aliquoted in 1 mg total protein aliquots until use.

Budding reactions were performed as described[40]. Briefly, 1 mg of microsomal membranes per 6–8 reactions was washed 3× with 2.5 M urea in B88 and 3× with B88. Budding reactions were set up in B88 to a final volume of 250 µl at the following concentrations: 10 µg/ml Sar1, 10 µg/ml Sec23/Sec24, 20 µg/ml Sec13/Sec31 and 0.1 mM nucleotide. Where appropriate, an ATP regeneration mix was included (final concentration 1 mM ATP, 50 µM GDP-mannose, 40 mM creatine phosphate, 200 µg/ml creatine phosphokinase). Reactions were incubated for 30 min at 25 °C and a 12 µl aliquot collected as the total fraction. The vesicle-containing supernatant was collected after pelleting the donor membrane (21,100 × g, 2 min, 4 °C). Vesicle fractions were then collected by centrifugation in a Beckman TLA-55 rotor (258,488 × g, 25 min, 4 °C). The supernatant was aspirated, the pelleted vesicles resuspended in SDS sample buffer and heated for 10 min at 55 °C with mixing. The samples were then analysed by SDS-PAGE and immunoblotting for Sec22 (Miller lab antibody) and Erv46 (a gift from Charles Barlowe). All experiments were repeated three times and a representative is shown.

**EM sample preparation**. For cryo-electron tomography: 4 µL of the GUV COPII BR was applied to negatively glow-discharged C-flat holey carbon coated gold grids (CF-4/1–4 AU, Electron Microscopy Sciences), blotted from both sides (60 s pre-blot wait, blot force setting five, and four second blot time) and plunge-frozen in 100% liquid ethane on a Vitrobot Mark IV (FEI) set to 4 °C and 100% humidity. 3 µL of BSA-blocked 5 nm gold nanoparticles (BBI Solutions) were added to a 30 µL GUV BR and gently agitated just prior to vitrification. Vitrified grids were stored in liquid nitrogen dewars to await data collection.

**For negative stain**. 4 µL of the GUV COPII BR was applied to negatively glow-discharged grids (Carbon film 300 Copper mesh, CF300-Cu), stained with 2% uranyl acetate, blotted with filter paper and air-dried at room temperature. Grids were imaged using either a Tecnai 120 keV TEM (T12) fitted with a CCD camera, or a Tecnai 200 keV TEM (F20) fitted with a DE20 detector (Direct Electron, San Diego). Unaligned and summed frames were collected for F20 images with a dose of 20–30 e−/pixel/s.

**Cryo-electron tomography data collection**. For wild-type COPII GUV BRs, a total of 286 dose-symmetric tilt series[42] with ±60° tilt range and 3° increments were acquired on Titan Krios operated at 300 keV in EFTEM mode with a Gatan Quantum energy filter (20 eV slit width) and K2 Summit direct electron detector (Gatan, Pleasanton CA) at ~1.33 Å/pixel. Data were collected at the ISMB EM facility in Birkbeck College and at the Cryo-EM Service Platform at EMBL Heidelberg. For ΔCTD-Sec31, 47 dose-symmetric tilt series were collected at Birkbeck with a K3 direct detector (Gatan, Pleasanton CA) at ~1.38 Å/pixel. For all sessions, defocus was systematically varied between 1.5 and 4.5 µm (Supplementary Table 2). Data were collected automatically using SerialEM[43] after manually selecting tubes using the AnchorMap procedure. Dose per tilt varied between 2.9 and 3.7 e−/Å², equating to ~120 and ~150 e−/Å² in total, respectively, depending on the dataset (Supplementary Table 2).

**Cryo-tomography data processing**. Tilt frames from Birkbeck were aligned using whole frame alignment with MotionCor2[44], which were amalgamated into ordered stacks. Tilt series were either aligned manually with IMOD or automatically with the Dynamo tilt series alignment (dtsa) pipeline[45]. Weighted back-projection was used to reconstruct bin8x tomograms with 50 iterations of SIRT-like filtering for initial particle-picking and STA. CTF estimation was performed with CTFFIND4 on a central rectangular region of the aligned and unbinned tilt series, as done previously[5]. The uncropped, aligned and unbinned tilt series were dose-weighted using critical exposure values determined previously using custom MATLAB scripts[46]. 3D-CTF correction and tomogram reconstruction was performed using the novaCTF pipeline[47], with bin2x, bin4x and bin8x versions calculated using IMOD binvol[48].

**Subtomogram averaging**. All STA and subsequent analysis was performed using a combination of Dynamo[45] and custom MATLAB scripts.

*Inner coat*. Initial particle-picking for the inner coat was performed using previously established protocols[5,15]. Briefly, tube axes were manually traced in IMOD to generate an oversampled lattice of cylindrical surface positions with angles assigned normal to the surface. 32³ voxel boxes were extracted from bin8x SIRT-like filtered tomograms for one round of initial reference-based alignments using a resampled and 50 Å low-pass filtered version of the inner coat reconstruction EMDB-0044[5]. Manual inspection of geometric markers using the UCSF Chimera placeObjects plug-in[49,50] confirmed convergence of oversampled coordinates onto the pseudo-helical inner coat lattice.

To rid outliers from the initial alignments, three strategies were used. Firstly, distance-based cleaning was applied using the Dynamo 'separation in tomogram' parameter, set to four pixels. This avoids duplication of data points by identifying clusters of converged particles and selecting the one with the highest cross-correlation (CC) score. Secondly, particles were cleaned based on their matching lattice directionality. Initial alignments were conducted on a tube-by-tube basis using the Dynamo in-plane flip setting to search in-plane rotation angles 180° apart. This allowed to assign directionality to each tube, and particles that were not conforming to it were discarded by using the Dynamo *dtgrep_direction* command in custom MATLAB scripts. Thirdly, manual CC-based thresholding was implemented to discard misaligned particles. As seen previously[5], particles on the tubule surface exhibited an orientation-dependent (Euler angle θ) CC score whereby top and bottom views had lower CC values. These were reweighted using the same polynomial fit for θ versus CC (MATLAB fit with option 'poly2') as described previously for more convenient thresholding. The cleaned initial coordinates were then combined and divided into half datasets for independent processing thereon.

Subsequent STA progressed through successive binning scales of 3DCTF-corrected tomograms, from bin8x to 4x, 2x then unbinned. At each level, angular and translational searches were reduced, with the low-pass filter determined by the Fourier shell correlation (FSC) 0.5 cut-off between the two half maps. A saddle-shaped mask mimicking the curvature of the membrane at the height of the inner coat layer was used throughout. A total of 151,176 particles contributed to the map.

The FSC between refined half maps reveals an average resolution of 4.6 Å at 0.143 cut-off. We note a sharp increase in the FSC in correspondence to the Nyquist frequency. We were unable to find the source of that increase, but we are confident that the resolution reported is correct as shown by the local resolution map (Supplementary Fig. 2c).

Half maps were used for density modification using Phenix[34]. The same mask used for alignments was imposed, and the density modification procedure was carried out without reducing the box size. All other parameters were used as default. After density modification, the map was further sharpened using the 'autosharpen' option in Phenix[51].

*Outer coat*. To target the sparser outer coat lattice for STA, we used the refined coordinates of the inner coat to locate the outer coat tetrameric vertices. Over-sampled coordinates for the outer coat were obtained by radially shifting refined inner coat coordinates by eight pixels further away from the membrane, following initial alignments from SIRT-like filtered tomograms (Supplementary Fig. 1b). 64³ voxel boxes were extracted from bin8x SIRT-like filtered tomograms for one round of initial reference-based alignments using a resampled and 50 Å low-pass filtered version of the outer coat tetrameric vertex reconstruction EMDB-2429[15]. Again, manual inspection of positions and orientations with the placeObject plug-in confirmed conformity to the expected lozenge-shaped outer coat lattice (Supplementary Fig. 1). Moreover, density for the neighbouring outer coat vertices emerged outside of the alignment mask and template, suggesting these initial alignments are not suffering from reference bias. The cleaned initial coordinates were then combined and divided into half datasets for independent processing thereon. The rhomboidal lattice can be appreciated by plotting the frequency of neighbouring particles for each vertex in the STA dataset (Supplementary Fig. 1d, left panel). Analysis of the relative positions between the inner coat and outer vertex did not reveal any defined spatial relationship (Supplementary Fig. 1d, right panel).

Subsequent STA progressed through successive binning scales of 3DCTF-corrected tomograms, from bin8x to 4x, to 2x, for a final pixel size of 2.654 Å. At each level, angular and translational searches were reduced, with the low-pass filter determined by the FSC 0.5 cut-off between the two half maps. A mask mimicking the curvature of the outer coat layer was used throughout. Prior to sharpening of the final unbinned map using relion 'postprocessing', the final half dataset averages were amplitude-weighted according to the sum of the combined CTFs.

The refined positions of vertices were used to extract two distinct datasets of left and right-handed rods respectively using the dynamo sub-boxing feature. Left-handed rods were processed as vertices, except that a cylindrical mask was used during alignments. Right-handed rods were subjected to classification in dynamo using multi-reference alignments. One class contained canonical rods, and particles belonging to this class were further processed as above. Two classes which contained the extra rod attachment were combined after applying a 180° in-plane rotation to particles in one class. After that, processing was carried out as for the other subtomograms.

The number of particles that contributed to outer coat averages is reported in Supplementary Table 3.

*Sec31-ΔCTD outer coat rods*. Oversampled coordinates for the outer coat were obtained in the same way as the WT dataset. Initial alignments using the previously resolved tetrameric vertex (EMDB-2429) did not produce lattice patterns conforming to the expected lozenge-shape as judged from the placeObjects inspection. This was confirmed by the subtomogram neighbour analysis for the refined initial alignment coordinates. Furthermore, the resulting average did not reveal new features emerging outside of the mask or initial reference. To confirm that Sec31-ΔCTD rods lack the appendage seen in the WT rod maps, we instead performed

initial alignments against a rod without handedness. For this, the final left-handed rod from the WT dataset was taken and rigid body fitted with the crystal structure (PDB 4bzk) in UCSF Chimera[49] to generate a 30 Å *molmap*. This was duplicated, mirror-symmetrised with the flipZ command in Chimera, and rotated along the axis of the rod by 180°, and summed with the original molmap using *vopMaximum* to create a Sec13/31 rod without handedness. This was used as a template for initial alignments, keeping Dynamo parameters consistent with vertex alignments at the same stage. This resulted in a rod which regained the original handedness of Sec13/31, suggesting no reference bias. To clean the dataset of misaligned particles, MRA with five classes and no shifts or rotations allowed was performed for 100 iterations. Two stable classes comprising ~90% of the data emerged as recognisable Sec13/31 rods. Refinement of each of these two selected classes against a wild-type left-handed rod gave averages that lacked the putative CTD appendage.

The procedure was repeated independently for two half datasets for resolution assessment.

*Subtomogram neighbour analysis.* To provide a semi-quantitative readout for the degree of lattice order, neighbour plots were calculated and used in a similar way to previous STA studies[52,53]. Briefly, all neighbouring particles are identified within a user-defined distance on a particle-by-particle basis. The relative orientation and distance to the matched particle is used to fill the relevant pixel in a master volume relative to its centre, which accumulates into a volume of integers. The final volumes are divided by the number of searched particles and normalised to a maximum intensity of one. For convenient visualisation, pixels in Z were summed to create heatmap representations (Supplementary Fig. 1d). This heatmap reflects the frequency of neighbouring particles and in a well-ordered lattice, peaks are visible. Furthermore, this master volume retains matched particle pairings, and can be masked to select specific relationships in the dataset (Supplementary Fig. 6c).

*Outer coat in spherical vesicles and cages.* Vesicles and cages were identified, and vertices manually picked from gaussian filtered binned x8 volumes using UCSF Chimera[49]. Initial orientations were assigned normal to the vesicle or cage centre, and the in-plane rotation angle was randomised. Vertices were then aligned for one iteration to the relevant starting reference (Supplementary Fig. 4), searching out of plane angles within a cone and the full in-plane rotation range.

**Fitting and interpretation.** The map output from phenix.resolve_cryoem[34] and the further sharpened map were used to provide guidance for model building.

Crystal structures of Sec23 (2QTV), Sec24 (1PCX), Sar1 (2QTV), Tango1 (5KYW) and Sec31 active peptide (2QTV) were fitted to the reconstruction using UCSF Chimera[49] and Coot[54].

Two copies of each protein were placed, representing two protomeric assemblies. Clashes between Sec23 from adjacent protomers were resolved by manual rebuilding. Clear density was also observed for residues 201–217 of Sec23, 363–371 and 463–466 of Sec24, and 157–159 of Sar1, and were manually built as they were absent from the crystal structures.

The model was refined with phenix.real_space_refine against the sharpened map, and validated with phenix.validation_cryoem (dev 3885) (Supplementary Table 4).

For the outer coat, the model of an entire rod (PDB 4bzj) was fitted as a rigid body into an initial map (as in Supplementary Fig. 1c, bottom panel) to obtain an initial position and orientation of each domain. These were then refined into the higher resolution 'focussed' maps by using the chimera 'fit in map' function.

**Homology modelling.** Sec31 residues 481-1273 (encompassing PRD and CTD regions) was used to search for remote homologues using the HHpred server[55], and identifying SRA1 as the closest homologue in the PDB database (PDB 2MGX, E-value 2.3E-15). To confirm the homology, the Sec31 protein sequence (Uniprot id: P38968) was used to search the CATH database of functional families[56], generating a significant hit to a Sec31 CTD functional family (E-value 6.0E-48). SRA1 matched this family with an *E*-value of 0.0001, which is within the threshold for homology modelling using functional family matches, based on previous benchmarks[57].

Homology models of Sec31 CTD were built using a combination of HHpred and Modeller[58], based on the highest-ranking homologue structure of SRA1[33]. According to calculations from proSA web server[59], the model has a *z*-score of −6.07, similar to that of the template (−5.38), and in line with that of all experimentally determined structures. Secondary structure and disorder predictions for Sec23 were performed using the PSIPRED server[60,61].

**Reporting summary.** Further information on research design is available in the Nature Research Reporting Summary linked to this article.

## Data availability

Data supporting the findings of this paper are available from the corresponding author upon reasonable request. A reporting summary for this Article is available as a Supplementary Information file. Source data are provided with this paper.

We have deposited the EM maps and models to the Electron Microscopy Data Bank with accession codes:

https://www.ebi.ac.uk/pdbe/entry/emdb/EMD-11193
https://www.ebi.ac.uk/pdbe/entry/emdb/EMD-11194
https://www.ebi.ac.uk/pdbe/entry/emdb/EMD-11197
https://www.ebi.ac.uk/pdbe/entry/emdb/EMD-11198
https://www.ebi.ac.uk/pdbe/entry/emdb/EMD-11199
https://www.ebi.ac.uk/pdbe/entry/emdb/EMD-11264
and to the Protein Data Bank with accession codes:
PDB 6ZG5 [https://doi.org/10.2210/pdb6ZG5/pdb]
PDB 6ZG6 [https://doi.org/10.2210/pdb6ZG6/pdb]
PDB 6ZGA [https://doi.org/10.2210/pdb6ZGA/pdb]
PDB 6ZL0 [https://doi.org/10.2210/pdb6ZL0/pdb]

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

## Acknowledgements

We thank Euan Pyle and Natasha Lukoyanova for comments on the paper. We thank Natasha Lukoyanova at the ISMB Birkbeck cryo-EM lab, and Wim Hagen at the EMBL cryo-EM facility in Heidelberg for cryo-tomography data collection, Tom Terwilliger at Los Alamos National Laboratory for advice on density modification procedures, and Christine Orengo and Natalie Dawson for help with Sec31 CTD bioinformatics analysis. This work was supported by grants from the Academy of Medical Sciences (Springboard award SBF0031030), the ERC (StG 852915—CRYTOCOP) and the BBSRC (BB/T002670/1) to G.Z., the UK Medical Research Council (MRC_UP_1201/10) to E.A.M. and the Wellcome Trust (102535/Z/13/Z) to A.C.M.C. (102535/Z/13/Z) and J.H. (Ph.D. studentship 109161/Z/15/A). Cryo-EM data for this investigation were collected at Birkbeck College, University of London, with financial support from the Wellcome Trust (202679/Z/16/Z and 206166/Z/17/Z), and at the Cryo-EM Service Platform at EMBL Heidelberg.

## Author contributions

Conceptualisation: G.Z., J.H.; Funding acquisition: G.Z., E.A.M.; Investigation: Sample preparation: J.H., N.R.B.; cryo-tomography data collection: J.H.; cryo-tomography and STA data processing: J.H., G.Z.; yeast viability and microsome budding assays: V.S., E.A.M.; model building and refinement: A.C. Writing: original draft: G.Z., J.H.; review and editing: all authors.

## Competing interests

The authors declare no competing interests.
