## [Peer Review File · Nature Communications]

Reviewer #1 (Remarks to the Author):

The report of J. Hutchings and colleagues is a structural study of the whole COPII complex polymerized on membranes by cryo-ET and subtomogram averaging.

The CopII machinery is involved in ER budding and vesicle formation for an anterograde traffic to the Golgi apparatus. Coatings are dynamic assemblies that assemble and dissociate at the ER. The COPII complex consists of 5 proteins, the GTPase Sar1 and the heterodimer Sec 23/Sec 24 form the internal coat while the heterotetramer Sec 13/Sec 31 forms the external coat. There are X-ray structures of Sec23, Sec24, Sar1 Sec31 active 683 peptide but no structure of the whole complex in the absence or presence of membrane. This limits our understanding of the mechanism of vesicle formation/assembly. The challenge of the structural analysis is that CopII is a multiprotein complex that forms variable supramolecular organization in the presence of membranes. The goal of the study is to describe the complete architecture of CopII to understand how CopII allows the formation of vesicles of different size and curvature.

This is a continuation of the work done by the same group that published in *Elife* 2013 and *Nat Com* 2018 with the same proteins and approaches by cryo-ET, subtomogram averaging and genetic of yeast. The in vitro system (yeast proteins, homologous expression, membrane system) are relevant. The cryo-ET and subtomogram approach is perfectly appropriate to answer the questions. The article is well written and the figures are well presented. The processing steps are very well described and performed with state of the art programs including a 3D CTF correction.

The article presents the molecular and supramolecular organization of the internal and external coats. The resolution is variable from 4.6 Å for the internal coat and 11-15 Å for the external coat. A protein organization is proposed by fitting the atomic models available in the EM envelope.

There is a very significant improvement in resolution on the outer coat from 40 Å to 11-15 Å (*elife* 2013) and a slight improvement of the inner coat from 4.9 Å to 4.6 Å resolution (*Nat Com* 2018). The gain in resolution allows to confirm the arrangement of all proteins in each coat and to determine areas of interaction between the coats which are then discussed as being involved in the flexibility of the complex and the recognition of curvatures. In vitro biochemical and genetic experiments in yeast provide confirmation of the data derived from cryo-EM analysis.

The novelties are more specifically:

1) A better resolution of sec 13/Sec 31 which allows a fit of sec 13/31 structures. By comparing the atomic model and the EM envelope, it is proposed a localization for an acid loop that would stabilize interactions with neighbors. To validate this proposal, biochemical and genetic experiments are done with sec31 deltaNTD. However, the deletion does not prevent tubule formation, which is interpreted as the fact that interactions within the outer coat are not necessary to generate curvature on deformable membrane. Figure supp 3E of the in vivo experiments is missing in my document. It is also difficult to see if the order of the inner coat is preserved in the negatively stained NTD sec31 tubes. A cryo-EM image or better a 3D reconstruction would be more convincing. Given the resolution in the proposed Nter region, probably less than 12 Å, the assignment of the Nter domain of sec31 seems plausible but not certain. This point is important because if the Nter is positioned elsewhere, the proposition that the outer coating is not needed for generating curvatures is less valid.

2) The organization of Beta propellers in the vertex is different from that reported in the absence of a membrane for human sec13/31 complexes. The analysis of yeast COPII vertex on vesicle excludes a difference related to curvature and attributes this organization to the species difference. This experiment is convincing.

3) Two new densities of the outer coat are observed and attributed to domains that allow interactions between outer and inner coat: a) a new density extra density attached to the rod

halfway between Sec13 and the dimerization interface. This density is attributed to the CTD domain of sec 13. A model of this domain is computed by homology with SRA1 and allows to propose a fit for the CTD domain in the EM envelop. However, the local resolution in this region of the EM does not allow a precise localization of the CTD domain. The lack of functional links between SAR1 and Sec31 questions the relevance of this modelisation. In vivo experiments confirmed the importance of this region for the stabilization of the external coat. b) additional and randomly distributed rods bridging between alpha-solenoid dimerization interfaces.

4) The internal coat is resolved to 4.6 Å resolution (4.7 Å written in the Table 3) which improves the current model at 4.9 Å, especially in the areas of interaction with the external coat. Thus known interaction regions are identified for the first time in the EM envelope: the region around the PPP sequence of sec31 which interacts with sec23 and a charged region of sec31 which interacts with sec23. In addition, a region of interaction between sec23 and sar1 is identified and validated by in vitro tubulation experiment.

Overall the study is very well performed from the cryo-ET and image analysis point of view. This allows to obtain a more complete model of the COPII complex than the current models and highlights interaction zones. This is an additional example of the advances in subtomogram averaging of flexible multiproteic complexes associated with in vitro membranes. The resolution, which is very good for the inner coat, remains average for the outer coat, which leaves ambiguities on the organization of the proteins.

This is an important advance on the structural organization of CopII complex. However, the impact on the understanding of vesicle formation and stabilization seems less impressive likely because the resolution of the outer coat is not sufficient for an unambiguous assignment of all sub-domains of proteins required for building a pseudo-atomic model. Biochemical and genetic experiments seem to confirm and complement the results of the cryo rather than providing original information. Thus, this report appears mostly as a structural study of a very good level and should be better considered in a journal such as Nature Structural and Cellular Biology. However, it could be considered for publication in Nature Communication after major modifications.

Major modifications

- The role of the NTD needs to be better characterized. The density is not clearly defined in the EM envelope. The resolution should be improved if possible in this area, the sec31 deltaNTD tubes should be analysis and compare to WT and new genetic or biochemical approaches should strength the proposed role of the NTD sub-domain.
- The CTD domain needs also to be better characterized. The density is defined in the EM envelope but the proposed model is questionable because of the medium resolution and the use of a protein with little homology to build a model.
- the EM volume of the outer coat has been built from ~15K sub-volumes (~150 K sub-volumes for the inner coat). It might be useful to present the 3D classes obtained during the processing of the outer coat and discuss the local differences between the 3D classes. This may allow a better understanding of the consequences of local flexible zones on the flexibility of the whole outer coat.
- A map with local resolution for the inner coat should be presented to highlight the flexibles regions in contact to the outer coat.

Minor points

In the introduction, a figure presenting the COPII proteins and their role would be useful for non-expert COPII readers.

There is no figure sup 3 E. "Anti-Sec31 Western Blot of the 5-FOA derived SEC31 and sec31-NTD strains showing that the 835 expected size of Sec31 variants is present in the surviving cells".

A representation of the EM envelope with the proposed fits for the inner and outer coat and membrane would be helpful to have an overview of the results (as depicted e.g. in figure 5 elife

2013).

Figure 7 is difficult to understand. A schematic 3D representation of the proteins and their interactions and with a lipid bilayer would be more useful.

Reviewer #2 (Remarks to the Author):

Structure of the complete, membrane-assembled COPII coat reveals a complex interaction network

In this study Hutchings and colleagues report the structure of the yeast COPII coat from an in vitro reconstituted system using purified proteins and GUVs. This work follows on from their 2018 paper where they used a similar system and methodology to study the COPII inner coat. This current study extends this and other published work to reveal the structure of the complete yeast COPII coat. While the resolutions of the structures determined do not allow for atomic detail to be discerned, but nevertheless the authors were able to verify known contacts and also identify some previously unknown contacts such as the Sec31 CTD. In general, the study is well conducted, rigorous and the authors have been careful not to overinterpret their conclusions beyond the resolutions obtained. These COPII structures and the information garnered from them will be a valuable resource to the intracellular trafficking field as well as cell and structural biologists in general, and I recommend publication provided the authors can address the following points -

Figure 1C – the fits appear ambiguous at the map contour level shown. Why would 180° rotated versions not fit equally well? Perhaps this part of the manuscript should be a bit more cautiously worded? Based on this, I might expect that a high-resolution structure might change the picture considerably in the future.

Figure 2B and line 172, the authors state: ‘This extra density is probably sub-stoichiometric, as we could see it clearly only at low contour levels (Fig. 2B, D) or in averages with lower sharpening levels (not shown). We reasoned that it could correspond to the Sec31 CTD, which is predicted to be a structured helical domain.’

I do not understand how the Sec31 CTD can be sub-stoichiometric? If the rest of Sec31 is present then the CTD must be as it is a domain of Sec31 and not an additional protein. Did the authors see degradation in their in vitro purifications? If so this should be stated. If not isn't the weakness of this density more likely to be due to flexibility of this particular region?

Figure 2B: Are the top and bottom images of panel 2B depicted at the same contour level? They seem different to me.

Figure 2D: Figure 2D is shown at a different contour level. The authors should define the contour levels used (i.e. sigma value) so the weakness of the density can be ascertained. This should be defined for all the panels in the manuscript showing EM density - could the authors please report the contour level in each panel please?

Figure S2B – there is a sharp increase in the FSC close to Nyquist frequency. Why is this? Needs to be corrected. Probably the resolution measurement will change as a result, which is fine.

Figure S3, Line 158: The differences seen between the human and yeast COPII structures at the vertices are potentially but the explanation given by the authors for this difference seems somewhat vague.

Is it possible for the authors to point to some structural/sequence element differences that could explain the difference in the structures or do the authors believe the differences are artefacts from differences in the reconstitution conditions?

In my opinion at 11-13 Å resolution not very informative to describe 15 Å RMSDs. Fits do not appear to me to be so clear cut unfortunately.

Line 210 -216: The authors discuss the tubulation ability of Sec31 mutants but the data is not shown. As this is the only data that supports their conclusions in lines 214-216, the data should be added to the supplement and the levels of tubulation seen quantified.

Figure 6C-E: The level of tubulation should be quantitated for the different mutants. At present it is unclear how drastic the effect of the mutations are.

Figure 6 C-E: Are all panels at the same scale? Could you add scale bars to all panels separately please?

Figure 6 D and E: Shouldn't the panels be labelled Sec31 ΔPPP?

Figures S3E and F -figure legend. The figure legend for these panels do not seem to be correct. Perhaps mixed up during submission, please correct.

There are also a few typos in the manuscript that should be corrected:

Line 94: there is an extra 'of'

Line 734: should be 'right panel'

Reviewer #3 (Remarks to the Author):

This is a very interesting new advance on the structure of the COPII coat complex that builds on work previously published by this group in Nature Communications in 2018. The new sample has yielded higher resolution and additional detail has become visible, enabling new hypotheses to be made regarding the roles of Sec 31/13 and 23/24 in coat assembly. This information is likely to be of interest not only to the membrane trafficking community but also to those interested in cellular protein assemblies in general. I would strongly recommend publication of this work in Nature Communications subject to clarification of the following points.

PDB validation report

The clashscore is relatively poor in comparison to other deposited EM structures (See red and blue slide bars in section 1). Could this be improved? I see that previously determined crystal structures

were fitted into the map so the clashes may be a legacy of these published models. However the methods state that the model was refined using Phenix and also mentioned some manual rebuilding. Could the authors comment on what aspects of their model are new or retained from the crystal structure data?

Also the PDB-calculated FSC curve differs from the author-provided FSC curve and suggests a lower overall map resolution. Could the authors comment on the reasons for this?

Manuscript

1. The abstract comments that the new information provided will 'significantly move away from the current paradigm'. It would be helpful to summarise what that paradigm is.

2. Line 679: Fitting and interpretation. The conclusions rely on the ability to accurately fit the relevant crystal structures into the map. It would be helpful to expand on the approach used for this. For example how did the authors decide how to arrange the individual sec 34/24 subunits before fitting? Was this based on previous work or were alternative arrangements tested?

3. Lines 685-687 'Clear density was also observed for residues 201-217 of Sec23, 363-371 and 463-466 of Sec24, and 157-159 of Sar1, and were manually built as they were absent from the crystal structures.' At 4.6 Ang the map is rather low resolution for model building. Some indication of the expected accuracy of the newly built sections of the structure should be provided. For example how confident are the authors of the register of the structure or the rotamer positions? It is certainly of value to fit a model where the density suggests this is possible, but the level of confidence that can be placed in that model should be more clearly stated. I am not certain that the phrase 'atomic' model is strictly appropriate here, valuable though the model proposed is.

4. Lines 157-8 In the comparison between vertex interactions between yeast and humans, it would be interesting to discuss the level of homology or otherwise between the sequences at this point.

Figures

Fig 1

A: I'm not completely clear what features the blue and red arrowheads are marking to indicate the inner and outer coats. Could this be clarified?

B: The legend to B refers to a panel E which is missing.

Supp Fig 3

A: Is the stub highlighted in green showing what is left of the acidic loop, or is it showing what is missing through comparison with another map? Also could the authors explain in more detail how the green density representing the proposed 339-357 negatively charged peptide was generated? Was this at the same contour level as the map density shown in grey?

C: I only see a very faint band for sec13 in the buoyant vesicle lane for full length sec31. Could the authors comment on this?

D: The ordered coat lattice is hard to see. Perhaps a magnified inset would help? Also it is not clear from the legend what the composition of the sample is. Can the authors clarify whether sec23 and 24 are present on the lipid in these experiments and therefore whether the tubulation is the result

of the delta sec31 binding to the sec23/24?

E: This panel is a structural figure not a western blot

Supp Fig 6

E: I'm not completely clear what the legend description means. Does this mean that two maps were superimposed to be able to see where the additional rods fit?

Supp Fig 7

C: The density for the bound nucleotide in C is not very well-defined. Could this be shown more clearly?

We would like to thank the reviewers for their insightful comments. We believe this has led us to produce an improved manuscript. Please find below our point-by-point response. We identify each response by the reviewer number and a letter for ease of cross-referencing.

Reviewer #1 (Remarks to the Author):

The report of J. Hutchings and colleagues is a structural study of the whole COPII complex polymerized on membranes by cryo-ET and subtomogram averaging.

The CopII machinery is involved in ER budding and vesicle formation for an anterograde traffic to the Golgi apparatus. Coatings are dynamic assemblies that assemble and dissociate at the ER. The COPII complex consists of 5 proteins, the GTPase Sar1 and the heterodimer Sec 23/Sec 24 form the internal coat while the heterotetramer Sec 13/Sec 31 forms the external coat. There are X-ray structures of Sec23, Sec24, Sar1 Sec31 active 683 peptide but no structure of the whole complex in the absence or presence of membrane. This limits our understanding of the mechanism of vesicle formation/assembly. The challenge of the structural analysis is that CopII is a multiprotein complex that forms variable supramolecular organization in the presence of membranes. The goal of the study is to describe the complete architecture of CopII to understand how CopII allows the formation of vesicles of different size and curvature.

This is a continuation of the work done by the same group that published in Elife 2013 and Nat Com 2018 with the same proteins and approaches by cryo-ET, subtomogram averaging and genetic of yeast. The in vitro system (yeast proteins, homologous expression, membrane system) are relevant. The cryo-ET and subtomogram approach is perfectly appropriate to answer the questions. The article is well written and the figures are well presented. The processing steps are very well described and performed with state of the art programs including a 3D CTF correction.

The article presents the molecular and supramolecular organization of the internal and external coats. The resolution is variable from 4.6 Å for the internal coat and 11-15 Å for the external coat. A protein organization is proposed by fitting the atomic models available in the EM envelope. There is a very significant improvement in resolution on the outer coat from 40 Å to 11-15 Å (Elife 2013) and a slight improvement of the inner coat from 4.9 Å to 4.6 Å resolution (Nat Com 2018). The gain in resolution allows to confirm the arrangement of all proteins in each coat and to determine areas of interaction between the coats which are then discussed as being involved in the flexibility of the complex and the recognition of curvatures. In vitro biochemical and genetic experiments in yeast provide confirmation of the data derived from cryo-EM analysis.

The novelties are more specifically:

1) A better resolution of sec 13/Sec 31 which allows a fit of sec 13/31 structures. By comparing the atomic model and the EM envelope, it is proposed a localization for an acid loop that would stabilize interactions with neighbors. To validate this proposal, biochemical and genetic experiments are done with sec31 deltaNTD. However, the deletion does not prevent tubule formation, which is interpreted as the fact that interactions within the outer coat are not necessary to generate curvature on deformable membrane. Figure supp 3E of the in vivo experiments is missing in my document. It is also difficult to see if the order of the inner coat is preserved in the negatively stained NTD sec31 tubes. A cryo-EM image or better a 3D reconstruction would be more convincing. Given the resolution in the proposed Nter region, probably less than 12 Å, the assignment of the Nter domain of sec31 seems plausible but not certain. This point is important because if the Nter is positioned elsewhere, the proposition that the outer coating is not needed for generating curvatures is less valid.

1A. We regret a misunderstanding: the experiments with deltaNTD were carried out to determine the physiological importance of the vertex interaction and cage assembly for membrane deformation, and not to validate the stabilisation provided by the negatively charged loop. We did indeed find that membrane deformation can be achieved in the absence of the Sec31 N-terminal domain mediating

outer coat vertex assembly, and that this is a condition that leads to productive budding in cells that have easily deformable ER membranes due to reduced cargo load.

We propose the membrane remodelling is driven by the inner coat, and in fact we can detect its ordered presence on tubules formed *in vitro*. The reviewer is correct in saying that the order is not immediately visible in the negatively stained images we present, so we have added an inset to Supplementary Figure 3D that shows the power spectrum of a typical coated tubule formed with Sec31 deltaNTD, where the inner coat lattice layer lines are indicated by arrows. This clearly shows inner coat order.

As for the uncertainty of the assignment of the N terminus to the densities at the vertex, in Figure 1 we report the vertex structure, which was obtained by focussing the refinements on a small region, and indeed the reviewer is right that from this structure it is not clear that the assignment is unambiguous. However, at earlier stages of the processing, we obtain averages that encompass neighbouring vertices and interconnecting rods in a single low-resolution map.

Given the fit of the rod-shaped X-ray model into these earlier maps, there is no ambiguity as to the identity of the domains at the vertex. We add here Figure R1 to illustrate this for the benefit of the reviewer. In addition, previous work utilising hydrogen-deuterium exchange has experimentally proved that the Sec31 NTD forms cage vertices (Noble et al, NSMB 2013).

Figure R1. Fit of full Sec13-31 heterotetramer into a low-resolution map that includes neighbouring vertices.

The X-ray model has a characteristic Z shape and unambiguously fits into the map, thereby defining the identity of each domain, including the Sec31 NTD beta-propeller as the protomers forming cage vertices.

Also, we apologise about panel S3E being missing, we had indeed removed it after internal revisions as we did not consider it essential, but forgot to remove the Figure legend. That panel contains a control experiment where we demonstrate that in *emp24* depleted cells that grow in the presence of Sec31 deltaNTD, this is not due to Sec31 reverting to wild type. Unless required by the editor or reviewer, we would leave that panel out.

2) The organization of beta propellers in the vertex is different from that reported in the absence of a membrane for human sec13/31 complexes. The analysis of yeast COPII vertex on vesicle excludes a difference related to curvature and attributes this organization to the species difference. This experiment is convincing.

3) Two new densities of the outer coat are observed and attributed to domains that allow interactions between outer and inner coat: a) a new density extra density attached to the rod halfway between Sec13 and the dimerization interface. This density is attributed to the CTD domain of sec 13. A model of this domain is computed by homology with SRA1 and allows to propose a fit for the CTD domain in the EM envelop. However, the local resolution in this region of the EM does not allow a precise localization of the CTD domain. The lack of functional links between SAR1 and Sec31 questions

the relevance of this modelisation. In vivo experiments confirmed the importance of this region for the stabilization of the external coat. b) additional and randomly distributed rods bridging between alpha-solenoid dimerization interfaces.

1B. We use Sra1 to build a homology model because we obtained significant similarity scores using HHpred searches. Moreover, we find that Sec31 CTD and Sar1 both belong to the same functional superfamily when searching the CATH database, and that the similarity between the respective subfamilies has an e-value which is widely considered beyond the threshold required for homology modelling. However, we agree with the reviewer that the functional and evolutionary link between these two proteins is unclear. To confirm the validity of our homology model we have used Robetta for ab initio structure prediction of the Sec31 CTD. When superimposed to the SRA1-derived model, the two structures are nearly identical, with a global RMSD of 2.3 Å, giving us confidence that the model we use is correct. We have modified the text as follows: “Since no atomic model for Sec31 CTD has been determined, we built a homology model to fit into the appendage density. Steroid Receptor RNA Activator protein (SRA1)³³ is a functionally unrelated protein that is found only in mammals, and its evolutionary links with Sec31 are unclear. Nevertheless, SRA1 and Sec31 CTD belong to the same evolutionary family and their similarity justifies the use of the SRA1 structure to build a homology model of the Sec31 CTD (see Methods). Rigid-body fitting the homology model in the appendage density shows consistency of size and features, although at this resolution we cannot determine the precise molecular interface (Supplementary Fig. 5A-C).”

4) The internal coat is resolved to 4.6 Å resolution (4.7 Å written in the Table 3) which improves the current model at 4.9 Å, especially in the areas of interaction with the external coat. Thus known interaction regions are identified for the first time in the EM envelope: the region around the PPP sequence of sec31 which interacts with sec23 and a charged region of sec31 which interacts with sec23. In addition, a region of interaction between sec23 and sar1 is identified and validated by in vitro tubulation experiment.

1C. We have corrected table 3.

Overall the study is very well performed from the cryo-ET and image analysis point of view. This allows to obtain a more complete model of the COPII complex than the current models and highlights interaction zones. This is an additional example of the advances in subtomogram averaging of flexible multiproteic complexes associated with in vitro membranes. The resolution, which is very good for the inner coat, remains average for the outer coat, which leaves ambiguities on the organization of the proteins.

This is an important advance on the structural organization of CopII complex. However, the impact on the understanding of vesicle formation and stabilization seems less impressive likely because the resolution of the outer coat is not sufficient for an unambiguous assignment of all sub-domains of proteins required for building a pseudo-atomic model. Biochemical and genetic experiments seem to confirm and complement the results of the cryo rather than providing original information. Thus, this report appears mostly as a structural study of a very good level and should be better considered in a journal such as Nature Structural and Cellular Biology. However, it could be considered for publication in Nature Communication after major modifications.

1D. We respectfully disagree with the reviewer in that the assignment of outer coat subunit is ambiguous. As we show above, we believe all domains that have been previously solved by X-ray crystallography can be assigned due to the peculiar rod-shape of the Sec13-31 complex. The one assignment that could not be made unambiguously from the fits (i.e. the Sec31 CTD) was validated by determination of the structure of a deletion mutant. We believe our biochemical and functional experiments add important insights, and do not just confirm the structural data. For example, the fact that either NTD or CTD deletion is lethal but rescued in emp24 delta background reveals that outer coat assembly functions to overcome membrane resistance rather than being the sole driver of membrane curvature. Also, our mutational analysis of the inner coat L-loop demonstrated how vesicle

shape is dictated by the interplay of inner and outer coat assembly. These insights could not be obtained by the structure alone.

Major modifications

- The role of the NTD needs to be better characterized. The density is not clearly defined in the EM envelope. The resolution should be improved if possible in this area, the sec31 deltaNTD tubes should be analysis and compare to WT and new genetic or biochemical approaches should strength the proposed role of the NTD sub-domain.

1E. We strongly believe that the resolution we obtained is enough to draw our conclusions on the assignment of the NTD in the EM map, we refer to point 1A above, including the newly added data on inner coat order in deltaNTD tubes. We're not sure what the reviewer would like to see regarding additional genetic and biochemical experiments. We already include EM analysis of GUV budding, microsome budding and cargo capture experiments, as well as viability assays in yeast. We also refer to our previous work using a N-his version of Sec31 which displays a weakened vertex interface, where the phenotype is more subtle, supporting cell growth and microsome budding but only in conditions where inner coat does not turn-over. We also show here that a functional vertex causes spherical vesicles to form when the inner coat lattice is weakened. All together we think these experiments clearly point to a role for the N-term as essential to reinforce the coat to overcome membrane resistance and drive towards spherical curvature in conditions where the inner coat can disassemble.

- The CTD domain needs also to be better characterized. The density is defined in the EM envelope but the proposed model is questionable because of the medium resolution and the use of a protein with little homology to build a model.

1F. We have responded to this comment above (see point 1B).

- the EM volume of the outer coat has been built from ~15K sub-volumes (~150 K sub-volumes for the inner coat). It might be useful to present the 3D classes obtained during the processing of the outer coat and discuss the local differences between the 3D classes. This may allow a better understanding of the consequences of local flexible zones on the flexibility of the whole outer coat.

1G. We regret the misunderstanding here. The various outer coat maps we present are not the result of classification, but of refining alignments of subtomograms extracted at different points in the tomogram. We performed classification only for the right-handed rods, as we show in Figure 4. We have added 'class 1' and 'class 2' to the Figure to distinguish the particular case where we did perform classification to the others, for which the picking procedure is explained in the Methods section.

- A map with local resolution for the inner coat should be presented to highlight the flexibles regions in contact to the outer coat.

1H. We agree this is useful to highlight the inner coat flexible regions. We now include this in Supplementary Figure 2C.

Minor points

In the introduction, a figure presenting the COPII proteins and their role would be useful for non-expert COPII readers.

1I. We have added an introductory panel to Figure 7 which provides a summary view of the coat components. We believe adding this to Figure 7 provides a useful overview for non-expert readers, and at the same time it improves the clarity of our model presented in Figure 7B.

There is no figure sup 3 E. "Anti-Sec31 Western Blot of the 5-FOA derived SEC31 and sec31-NTD strains showing that the 835 expected size of Sec31 variants is present in the surviving cells".

1J. We apologise for the mistake. We have now removed this from the Figure legend (see also point 1A).

A representation of the EM envelope with the proposed fits for the inner and outer coat and membrane would be helpful to have an overview of the results (as depicted e.g. in figure 5 elife 2013).

1K. We already have figures with the models fitted into the cryo-EM maps. Due to the flexibility of each region with respect to the others, it is not possible to produce a map representing all coat layers simultaneously, and we feel the best way to represent them together is through a schematic representation, as in Figure 7.

Figure 7 is difficult to understand. A schematic 3D representation of the proteins and their interactions and with a lipid bilayer would be more useful.

1L. We feel adding a third dimension to the schematic of the protein and their interaction would make it even harder to understand. To improve clarity, we have added the membrane for reference and moved the labels next to the relevant proteins. We also added a panel (A) to provide an overview of the components (see point 1I), as well as summarising our findings on the function of outer cot assembly. We hope this improves the figure.

Reviewer #2 (Remarks to the Author):

Structure of the complete, membrane-assembled COPII coat reveals a complex interaction network

In this study Hutchings and colleagues report the structure of the yeast COPII coat from an in vitro reconstituted system using purified proteins and GUVs. This work follows on from their 2018 paper where they used a similar system and methodology to study the COPII inner coat. This current study extends this and other published work to reveal the structure of the complete yeast COPII coat. While the resolutions of the structures determined do not allow for atomic detail to be discerned, but nevertheless the authors were able to verify known contacts and also identify some previously unknown contacts such as the Sec31 CTD. In general, the study is well conducted, rigorous and the authors have been careful not to overinterpret their conclusions beyond the resolutions obtained. These COPII structures and the information garnered from them will be a valuable resource to the intracellular trafficking field as well as cell and structural biologists in general, and I recommend publication provided the authors can address the following points –

Figure 1C – the fits appear ambiguous at the map contour level shown. Why would 180° rotated versions not fit equally well? Perhaps this part of the manuscript should be a bit more cautiously worded? Based on this, I might expect that a high-resolution structure might change the picture considerably in the future.

2A. We have addressed this point in our response to reviewer 1A. The overall orientation of each domain is given by the fit of the rod-shaped X-ray model into the low-resolution density (see figure R1 above). We subsequently refined the rigid-body fits into the higher-resolution density, aided by the well defined ‘flower-like’ shape of the beta propeller, but did not change the overall orientation.

Figure 2B and line 172, the authors state: ‘This extra density is probably sub-stoichiometric, as we could see it clearly only at low contour levels (Fig. 2B, D) or in averages with lower sharpening levels (not shown). We reasoned that it could correspond to the Sec31 CTD, which is predicted to be a structured helical domain.’

I do not understand how the Sec31 CTD can be sub-stoichiometric? If the rest of Sec31 is present then the CTD must be as it is a domain of Sec31 and not an additional protein. Did the authors see

degradation in their in vitro purifications? If so this should be stated. If not isn't the weakness of this density more likely to be due to flexibility of this particular region?

2B. The reviewer raises a very good point here. We expect the CTD is present in most Sec31 molecules in this sample. In our protein preparations some degradation is always detected, probably due to the presence of the flexible linker, but the vast majority of protein remains full-length. We have added Coomassie-stained gel of purified Sec13-31 together with all the other components used in our structural analyses (Supplementary Figure 1E). What we propose in the original manuscript is that the CTD might be bound to the rods in sub-stoichiometric amounts. That is, occupancy along the rods might not be 100% even though the CTD itself is present at near-stoichiometric amounts. Given that the CTD is anchored by a flexible linker that reaches to the inner coat and then back to the outer coat (as schematised in Fig. 7B), it is possible that some of these domains are just 'floating' without being bound to the Sec31 rods. The most likely explanation is a combination of all: disorder, unbound states, and degradation.

In addition to the new Supplementary Figure 1E panel, we changed the manuscript to "The size of this appendage is indicative of a full domain. We reasoned that it could correspond to the Sec31 CTD, which is predicted to be a structured helical domain^{27,28}. We could see this extra density clearly only at low contour levels (Fig. 2B,D) or in averages with lower sharpening levels (not shown), indicating either flexibility or sub-stoichiometric binding, which could be a consequence of some domains not being bound, or missing due to degradation (Supplementary Fig 1E)."

Figure 2B: Are the top and bottom images of panel 2B depicted at the same contour level? They seem different to me.

2C. They are depicted at the same contour, as now specified in the legend.

Figure 2D: Figure 2D is shown at a different contour level. The authors should define the contour levels used (i.e. sigma value) so the weakness of the density can be ascertained. This should be defined for all the panels in the manuscript showing EM density - could the authors please report the contour level in each panel please?

2D. The reviewer is right that we have depicted the density in panel D at different contour levels, to show the weaker density for the CTD. We have now added the sigma threshold level throughout the figures for each panel.

Figure S2B – there is a sharp increase in the FSC close to Nyquist frequency. Why is this? Needs to be corrected. Probably the resolution measurement will change as a result, which is fine.

2E. The high FSC value is seen only at the very last pixel of the FSC. We were not able to identify the cause of that increase at Nyquist. It is present in many sub-tomogram averaging studies from other groups as well (we can provide references if required). We do not believe this affects the resolution we report for our average structure, for a number of reasons:

- the FSC descends to zero and remains around zero until Nyquist.
- the FSC we show is calculated with relion which weights for the effect of the mask. We add the full output from relion for the benefit of the reviewer in Figure R2. The curves corresponding to the unmasked map (green), the phase-randomised map (red), and the masked maps (blue) appear free of artifacts aside of the reported high value at Nyquist. The corrected curve (black) is the one we report in Supplementary Fig. 2.
- our average resolution matches well the local resolution distribution which we now report in Supplementary Figure 2C.

Figure S3, Line 158: The differences seen between the human and yeast COPII structures at the vertices are potentially but the explanation given by the authors for this difference seems somewhat vague.

Is it possible for the authors to point to some structural/sequence element differences that could explain the difference in the structures or do the authors believe the differences are artefacts from differences in the reconstitution conditions?

In my opinion at 11-13 Å resolution not very informative to describe 15 Å RMSDs. Fits do not appear to me to be so clear cut unfortunately.

2F. We did not detect any striking pattern that relates homology between human and yeast Sec31, and their common and different contact points at vertices. We include here Figure R3 where show this. This does not exclude that some significant homology or variability is present at the specific contact sites, but due to the intermediate resolution of our vertex map we do not feel confident making any residue-level claims. For this reason we would prefer not to include Figure R3 in the manuscript, but are ready to do so if the reviewer think this is necessary, together with a discussion regarding the uncertainty of any interpretation. We cannot exclude that the difference is due to reconstitution conditions, and future studies on human proteins coating membranes should clarify this.

We are confident that the differences between the two arrangements we see are genuine, even if the resolution of the map is ~12 Å. The centroid of the beta-propeller domains in the two different vertex structures can be placed accurately based on the fit of full rod models both in ours and in Stagg's cage structure (see also point 1A). The distance between the centroids can also be inferred – independently of the maps - from the analysis of the distance between neighbouring vertices running along the left or right-handed directions. We measure distances between vertices of 289 and 303 Å along the right and left-handed direction respectively. Within our stated resolution of ~12 Å, these measurements are compatible with the distance predicted by our model (292 and 299 Å respectively), but not with the Stagg model (274 and 314 Å).

Figure R3. Sec31 sequence alignments.

- a. Conservation between Sec31A from 9 different species, including *S. cerevisiae*. The N-terminal beta-propeller domain in highlighted within a red box.
- b. Details of the alignment for Sec31 NTD, including sequence, and conservation, quality, consensus and occupancy scores. Green and blue column indicate regions that face the interface between protomers in our vertex arrangement, and do not reveal any striking pattern.
- c. Structure of the fitted protomers, with the residue close to the interface highlighted in green and blue, as in panel b.

Line 210 -216: The authors discuss the tubulation ability of Sec31 mutants but the data is not shown. As this is the only data that supports their conclusions in lines 214-216, the data should be added to the supplement and the levels of tubulation seen quantified.

2G. The reason we are not showing the data is that there was no tubulation, so we would only show images of naked GUVs, which we think would be misleading as there are always some naked GUVs also in functional budding reactions. In this case, we saw no tubes at all in all three independent repeats, therefore we feel confident in saying that the mutant is not functional.

Figure 6C-E: The level of tubulation should be quantitated for the different mutants. At present it is unclear how drastic the effect of the mutations are.

2H. The reviewer raises an important point. We have attempted to quantify tubulation, as we agree it would be a very useful way to assess efficiency of each mutant. Unfortunately, it is difficult to quantify tubulation due to the fact that the number and size of GUVs is different in each reaction. Because of this, we cannot comfortably 'count' tubules or their density in a comparative manner. We think that budding from microsomal membranes and quantification of cargo is a more reliable method to quantify relative efficiency of mutants. In Figure 6 C-E we aim to convey not a difference in efficiency but a striking effect on vesicle morphology.

Figure 6 C-E: Are all panels at the same scale? Could you add scale bars to all panels separately please?

2I. Yes all panels are at the same scale. For clarity, we have added the scale bar to each separately.

Figure 6 D and E: Shouldn't the panels be labelled Sec31 Δ PPP?

2J. Yes, we thank the reviewer for spotting this mistake. We have now changed this.

Figures S3E and F -figure legend. The figure legend for these panels do not seem to be correct. Perhaps mixed up during submission, please correct.

2K. Done

There are also a few typos in the manuscript that should be corrected:

Line 94: there is an extra 'of'

Line 734: should be 'right panel'

2L. Done

Reviewer #3 (Remarks to the Author):

This is a very interesting new advance on the structure of the COPII coat complex that builds on work previously published by this group in Nature Communications in 2018. The new sample has yielded higher resolution and additional detail has become visible, enabling new hypotheses to be made regarding the roles of Sec 31/13 and 23/24 in coat assembly. This information is likely to be of interest not only to the membrane trafficking community but also to those interested in cellular protein assemblies in general. I would strongly recommend publication of this work in Nature Communications subject to clarification of the following points.

PDB validation report

The clashscore is relatively poor in comparison to other deposited EM structures (See red and blue slide bars in section 1). Could this be improved? I see that previously determined crystal structures were fitted into the map so the clashes may be a legacy of these published models. However the methods state that the model was refined using Phenix and also mentioned some manual rebuilding. Could the authors comment on what aspects of their model are new or retained from the crystal structure data?

3A. The clashscore is 13.2, which, according to molprobity, is in the 57th percentile for all structures and in the 97th for structures of similar resolutions. We would argue therefore that the clashscore is good given our map's resolution. The reviewer is right in that the model was refined from crystal structures previously obtained by other groups, which had worse clashscore (17), despite being at higher resolution. During refinement we have chosen restraint levels appropriate to the resolution of our map (Afonine et al., Acta Cryst D 2018). In regards to which part were rebuilt and which were taken from previous X-ray data, we refer to the following, from materials and methods: 'Clear density was also observed for residues 201-217 of Sec23, 363-371 and 463-466 of Sec24, and 157-159 of Sar1,

and were manually built as they were absent from the crystal structures.’ Aside from the residues mentioned here, all the others were refined starting from the X-ray structures.

Also the PDB-calculated FSC curve differs from the author-provided FSC curve and suggests a lower overall map resolution. Could the authors comment on the reasons for this?

3B. For full transparency, we have deposited to EMDDB the maps masked only with a box-sized soft spherical mask (which we refer to as unmasked), and have deposited the masks we used for the FSC calculation we present in the paper separately. The PDB validation automatically calculates the FSC between half-maps without applying any mask, which is why they appear different from our Figure. The mask we use for FSC calculation encompasses the central, better resolved, subunit. The PDB-calculated FSC includes the neighbouring subunits all the way to the box limits, thereby including lower resolution regions in the average FSC. We refer for this to our response 2E (Figure R2), where we show the FSC of unmasked and masked maps in their original plots from relion, and to our point 1H, where we now include the local resolution map in Supplementary Figure 2.

Manuscript

1. The abstract comments that the new information provided will ‘significantly move away from the current paradigm’. It would be helpful to summarise what that paradigm is.

3C. We have now modified the abstract to more explicitly explain this. We have also added a panel to Figure 7 where we focus on the shift in paradigm, depicting the ability of a coat whose outer coat cannot form cages to promote membrane deformation in the absence of cargo.

2. Line 679: Fitting and interpretation. The conclusions rely on the ability to accurately fit the relevant crystal structures into the map. It would be helpful to expand on the approach used for this. For example how did the authors decide how to arrange the individual sec 34/24 subunits before fitting? Was this based on previous work or were alternative arrangements tested?

3D. The fit of Sar1-Sec23-24 was indeed based on previous work, but even without access to previously obtained maps, it would have been unambiguous to assign these subunits at the resolution of 4.6 Å.

As for the outer coat, we refer to our first response point 1A, where we explain that assignment of domains and their overall orientation was done based on the very characteristic shape of Sec13-31 at the initial stages of alignment, where the entire rod and adjacent vertices are all visible in the same map at low resolution. Indeed, we realised we did not explain how we fitted outer coat components in the methods, and we have now added: “For the outer coat, the model of an entire rod (PDB 4bzj) was fitted as a rigid body into an initial map (as in Supplementary Figure 1C, bottom panel) to obtain an initial position and orientation of each domain. These were then refined into the higher resolution ‘focussed’ maps by using the chimera ‘fit in map’ function.”

3. Lines 685-687 ‘Clear density was also observed for residues 201-217 of Sec23, 363-371 and 463-466 of Sec24, and 157-159 of Sar1, and were manually built as they were absent from the crystal structures.’ At 4.6 Å the map is rather low resolution for model building. Some indication of the expected accuracy of the newly built sections of the structure should be provided. For example how confident are the authors of the register of the structure or the rotamer positions? It is certainly of value to fit a model where the density suggests this is possible, but the level of confidence that can be placed in that model should be more clearly stated.

3E. We agree with the reviewer that model building at 4.6 Å can be inaccurate. We performed de novo building only in missing regions that are short but traceable. We are reasonably confident about their register, given their limited length and the fact that the end points of the newly built regions are well defined. As for the longer, 17AA region of Sec23 (residues 201-217, named as the ‘L-loop’): the corresponding density is clearly defined (being ‘sausage-like’) and the N- and C-termini are in close

proximity, such that we were only able to build it as an alpha helix (Figure 6a and b). Nevertheless, we cannot be absolutely certain about rotamer positions and generally deferred to library conformations. Hence we have removed the word 'atomic' in the manuscript when referring to refined or fitted models, and are prepared to truncate newly built side chains from our model if required.

4. Lines 157-8 In the comparison between vertex interactions between yeast and humans, it would be interesting to discuss the level of homology or otherwise between the sequences at this point.

3F. We agree that this would be very interesting, and we refer to our response to reviewer 2F.

Figures

Fig 1

A: I'm not completely clear what features the blue and red arrowheads are marking to indicate the inner and outer coats. Could this be clarified?

3G. We have clarified this in the figure legend which now reads: "A. Slices through different z heights of a binned and filtered representative cryo-tomogram of wild type COPII-coated tubule. In the top panel, density for the inner coat lattice is visible from the top and is indicated by a blue arrow, while the outer coat is cut through its side (red arrow). In the bottom panel, the outer coat is visible from the top, with its characteristic lozenge pattern (red arrow). Scale bar 50 nm."

B: The legend to B refers to a panel E which is missing.

3H. We have now rectified this error.

Supp Fig 3

A: Is the stub highlighted in green showing what is left of the acidic loop, or is it showing what is missing through comparison with another map? Also could the authors explain in more detail how the green density representing the proposed 339-357 negatively charged peptide was generated? Was this at the same contour level as the map density shown in grey?

3I. The green density is indeed a difference map between our subtomogram average and the X-ray model (which is missing the disordered acidic loop), and we propose that density might correspond to the acidic loop that becomes ordered in the context of the assembled vertex. We have coloured the residues in the X-ray model that flank the missing loop in green, which is what we call 'stub'. We realise this is not very clear, and we have modified the legend: 'A bottom view of the vertex reconstruction (outlined transparent white), with fitted models (Sec31 in dark red and orange, and Sec13 in grey). In the model, the N-terminal residues are coloured in blue, and residues that flank an acidic loop missing from the X-ray structures are coloured in green and indicated by a box with its aminoacidic sequence. The difference map between the subtomogram average and the model is shown in green, and is consistent with being occupied by the acidic loop.'

C: I only see a very faint band for sec13 in the buoyant vesicle lane for full length sec31. Could the authors comment on this?

3J. The Sec13 signal is consistently less strong than Sec31, due to its small size. However, the ratio of Sec13 to Sec31 is the same in the floated samples and input lanes. In the case of the full-length Sec31, there is less protein floating overall but both samples were normalized to account for recovery of lipids at the top of the gradient to ensure equivalent loading relative to liposome recovery.

D: The ordered coat lattice is hard to see. Perhaps a magnified inset would help? Also it is not clear from the legend what the composition of the sample is. Can the authors clarify whether sec23 and 24

are present on the lipid in these experiments and therefore whether the tubulation is the result of the delta sec31 binding to the sec23/24?

3K. We have added an inset with the power spectrum of the ordered inner coat lattice, which shows clear layer lines. We have also modified the figure legend which now reads: "Negatively stained GUV budding reconstitutions performed with the full set of COPII proteins, but where Sec31 was deleted of its N-terminal β -propeller domain. These show tubulation and ordered inner coat lattice."

E: This panel is a structural figure not a western blot

3L. We have now rectified this error, as explained in our previous responses to reviewers 1A and 2K.

Supp Fig 6

E: I'm not completely clear what the legend description means. Does this mean that two maps were superimposed to be able to see where the additional rods fit?

3M. We have now clarified to: "E. Rods selected in D (green) were placed in their original positions and orientation in a representative tomogram, together with the canonical rods and vertices (dark pink and red, respectively), showing that they bridge between patches of mismatched outer coat lattice."

Supp Fig 7

C: The density for the bound nucleotide in C is not very well-defined. Could this be shown more clearly?

3N. We have substituted the panel with a similar one, where the nucleotide is coloured in red and the density around it was darkened to improve its visibility. The purpose of that panel was to show that the features we see in the core of Sar1 (nucleotide and beta-strand separation) match the expectation of a 4.6 Å map, and we hope this change helps convey our message.

Reviewer #1 (Remarks to the Author):

The authors have answered to my questions and modified the manuscript in consequence. below are my new comments- starting by Reviewer 1:

Reviewer #1 (Remarks to the Author):

The report of J. Hutchings and colleagues is a structural study of the whole COPII complex polymerized on membranes by cryo-ET and subtomogram averaging. The CopII machinery is involved in ER budding and vesicle formation for an anterograde traffic to the Golgi apparatus. Coatings are dynamic assemblies that assemble and dissociate at the ER. The COPII complex consists of 5 proteins, the GTPase Sar1 and the heterodimer Sec 23/Sec 24 form the internal coat while the heterotetramer Sec 13/Sec 31 forms the external coat. There are X-ray structures of Sec23, Sec24, Sar1 Sec31 active 683 peptide but no structure of the whole complex in the absence or presence of membrane. This limits our understanding of the mechanism of vesicle formation/assembly. The challenge of the structural analysis is that CopII is a multiprotein complex that forms variable supramolecular organization in the presence of membranes. The goal of the study is to describe the complete architecture of CopII to understand how CopII allows the formation of vesicles of different size and curvature.

This is a continuation of the work done by the same group that published in Elife 2013 and Nat Com 2018 with the same proteins and approaches by cryo-ET, subtomogram averaging and genetic of yeast. The in vitro system (yeast proteins, homologous expression, membrane system) are relevant. The cryo-ET and subtomogram approach is perfectly appropriate to answer the questions. The article is well written and the figures are well presented. The processing steps are very well described and performed with state of the art programs including a 3D CTF correction.

The article presents the molecular and supramolecular organization of the internal and external coats. The resolution is variable from 4.6 Å for the internal coat and 11-15 Å for the external coat. A protein organization is proposed by fitting the atomic models available in the EM envelope. There is a very significant improvement in resolution on the outer coat from 40 Å to 11-15 Å (elife 2013) and a slight improvement of the inner coat from 4.9 Å to 4.6 Å resolution (Nat Com 2018). The gain in resolution allows to confirm the arrangement of all proteins in each coat and to determine areas of interaction between the coats which are then discussed as being involved in the flexibility of the complex and the recognition of curvatures. In vitro biochemical and genetic experiments in yeast provide confirmation of the data derived from cryo-EM analysis.

The novelties are more specifically:

- 1) A better resolution of sec 13/Sec 31 which allows a fit of sec 13/31 structures. By comparing the atomic model and the EM envelope, it is proposed a localization for an acid loop that would stabilize interactions with neighbors. To validate this proposal, biochemical and genetic experiments are done with sec31 deltaNTD. However, the deletion does not prevent tubule formation, which is interpreted as the fact that interactions within the outer coat are not necessary to generate curvature on deformable membrane. Figure supp 3E of the in vivo experiments is missing in my document. It is also difficult to see if the order of the inner coat is preserved in the negatively stained NTD sec31 tubes. A cryo-EM image or better a 3D reconstruction would be more convincing. Given the resolution in the proposed Nter region, probably less than 12 Å, the assignment of the Nter domain of sec31 seems plausible but not certain. This point is important because if the Nter is positioned elsewhere, the proposition that the outer coating is not needed for generating curvatures is less valid.

1A. We regret a misunderstanding: the experiments with deltaNTD were carried out to determine the physiological importance of the vertex interaction and cage assembly for membrane deformation, and not to validate the stabilisation provided by the negatively charged loop. We did indeed find that membrane deformation can be achieved in the absence of the Sec31 N-terminal domain mediating outer coat vertex assembly, and that this is a condition that leads to productive budding in cells that have easily deformable ER membranes due to reduced cargo load.

We propose the membrane remodelling is driven by the inner coat, and in fact we can detect its ordered presence on tubules formed in vitro. The reviewer is correct in saying that the order is not immediately visible in the negatively stained images we present, so we have added an inset to Supplementary Figure 3D that shows the power spectrum of a typical coated tubule formed with Sec31 deltaNTD, where the inner coat lattice layer lines are indicated by arrows. This clearly shows inner coat order.

Reviewer 1: I agree with the explanations on the role of deltaNTD and the modification of the figure

As for the uncertainty of the assignment of the N terminus to the densities at the vertex, in Figure 1 we report the vertex structure, which was obtained by focussing the refinements on a small region, and indeed the reviewer is right that from this structure it is not clear that the assignment is unambiguous. However, at earlier stages of the processing, we obtain averages that encompass neighbouring vertices and interconnecting rods in a single low-resolution map.

Given the fit of the rod-shaped X-ray model into these earlier maps, there is no ambiguity as to the identity of the domains at the vertex. We add here Figure R1 to illustrate this for the benefit of the reviewer. In addition, previous work utilising hydrogen-deuterium exchange has experimentally proved that the Sec31 NTD forms cage vertices (Noble et al, NSMB 2013).

Reviewer 1: I agree with the explanations. If possible to add the figure R1 in a supplementary figure, it could be useful for the reader.

Also, we apologise about panel S3E being missing, we had indeed removed it after internal revisions as we did not consider it essential, but forgot to remove the Figure legend. That panel contains a control experiment where we demonstrate that in emp24 depleted cells that grow in the presence of Sec31 deltaNTD, this is not due to Sec31 reverting to wild type. Unless required by the editor or reviewer, we would leave that panel out.

2) The organization of beta propellers in the vertex is different from that reported in the absence of a membrane for human sec13/31 complexes. The analysis of yeast COPII vertex on vesicle excludes a difference related to curvature and attributes this organization to the species difference. This experiment is convincing.

3) Two new densities of the outer coat are observed and attributed to domains that allow interactions between outer and inner coat: a) a new density extra density attached to the rod halfway between Sec13 and the dimerization interface. This density is attributed to the CTD domain of sec 13. A model of this domain is computed by homology with SRA1 and allows to propose a fit for the CTD domain in the EM envelop. However, the local resolution in this region of the EM does not allow a precise localization of the CTD domain. The lack of functional links between SAR1 and Sec31 questions the relevance of this modelisation. In vivo experiments confirmed the importance of this region for the stabilization of the external coat. b) additional and randomly distributed rods

bridging between alpha-solenoid dimerization interfaces.

1B. We use Sra1 to build a homology model because we obtained significant similarity scores using HHpred searches. Moreover, we find that Sec31 CTD and Sar1 both belong to the same functional superfamily when searching the CATH database, and that the similarity between the respective subfamilies has an e-value which is widely considered beyond the threshold required for homology modelling. However, we agree with the reviewer that the functional and evolutionary link between these two proteins is unclear. To confirm the validity of our homology model we have used Robetta for ab initio structure prediction of the Sec31 CTD. When superimposed to the SRA1-derived model, the two structures are nearly identical, with a global RMSD of 2.3 Å, giving us confidence that the model we use is correct. We have modified the text as follows: "Since no atomic model for Sec31 CTD has been determined, we built a homology model to fit into the appendage density. Steroid Receptor RNA Activator protein (SRA1) 33 is a functionally unrelated protein that is found only in mammals, and its evolutionary links with Sec31 are unclear. Nevertheless, SRA1 and Sec31 CTD belong to the same evolutionary family and their similarity justifies the use of the SRA1 structure to build a homology model of the Sec31 CTD (see Methods). Rigid-body fitting the homology model in the appendage density shows consistency of size and features, although at this resolution we cannot determine the precise molecular interface (Supplementary Fig. 5A-C)."

Reviewer 1: The additional details in the Methods are important for the understanding of the strategy used for the modelisation of Sec31-NTD. Could you add if the homology model has been built using the mode ab initio, homology or deep learning TrRosetta or Robetta as the confidence value. Future works will definitively answer the validity of this prediction.

4) The internal coat is resolved to 4.6 Å resolution (4.7 Å written in the Table 3) which improves the current model at 4.9 Å, especially in the areas of interaction with the external coat. Thus known interaction regions are identified for the first time in the EM envelope: the region around the PPP sequence of sec31 which interacts with sec23 and a charged region of sec31 which interacts with sec23. In addition, a region of interaction between sec23 and sar1 is identified and validated by in vitro tubulation experiment.

1C. We have corrected table 3.

Overall the study is very well performed from the cryo-ET and image analysis point of view. This allows to obtain a more complete model of the COPII complex than the current models and highlights interaction zones. This is an additional example of the advances in subtomogram averaging of flexible multiproteic complexes associated with in vitro membranes. The resolution, which is very good for the inner coat, remains average for the outer coat, which leaves ambiguities on the organization of the proteins.

This is an important advance on the structural organization of CopII complex. However, the impact on the understanding of vesicle formation and stabilization seems less impressive likely because the resolution of the outer coat is not sufficient for an unambiguous assignment of all sub-domains of proteins required for building a pseudo-atomic model. Biochemical and genetic experiments seem to confirm and complement the results of the cryo rather than providing original information. Thus, this report appears mostly as a structural study of a very good level and should be better considered in a journal such as Nature Structural and Cellular Biology. However, it could be considered for publication in Nature Communication after major modifications.

1D. We respectfully disagree with the reviewer in that the assignment of outer coat subunit is ambiguous. As we show above, we believe all domains that have been previously solved by X-ray crystallography can be assigned due to the peculiar rod-shape of the Sec13-31 complex. The one

assignment that could not be made unambiguously from the fits (i.e. the Sec31 CTD) was validated by determination of the structure of a deletion mutant. We believe our biochemical and functional experiments add important insights, and do not just confirm the structural data. For example, the fact that either NTD or CTD deletion is lethal but rescued in emp24 delta background reveals that outer coat assembly functions to overcome membrane resistance rather than being the sole driver of membrane curvature. Also, our mutational analysis of the inner coat L-loop demonstrated how vesicle shape is dictated by the interplay of inner and outer coat assembly. These insights could not be obtained by the structure alone.

Reviewer 1: I agree with the explanations.

Major modifications

- The role of the NTD needs to be better characterized. The density is not clearly defined in the EM envelope. The resolution should be improved if possible in this area, the sec31 deltaNTD tubes should be analysis and compare to WT and new genetic or biochemical approaches should strength the proposed role of the NTD sub-domain.

1E. We strongly believe that the resolution we obtained is enough to draw our conclusions on the assignment of the NTD in the EM map, we refer to point 1A above, including the newly added data on inner coat order in deltaNTD tubes. We're not sure what the reviewer would like to see regarding additional genetic and biochemical experiments. We already include EM analysis of GUV budding, microsome budding and cargo capture experiments, as well as viability assays in yeast. We also refer to our previous work using a N-his version of Sec31 which displays a weakened vertex interface, where the phenotype is more subtle, supporting cell growth and microsome budding but only in conditions where inner coat does not turn-over. We also show here that a functional vertex causes spherical vesicles to form when the inner coat lattice is weakened. All together we think these experiments clearly point to a role for the N-term as essential to reinforce the coat to overcome membrane resistance and drive towards spherical curvature in conditions where the inner coat can disassemble.

Reviewer 1: I agree with the explanations.

- The CTD domain needs also to be better characterized. The density is defined in the EM envelope but the proposed model is questionable because of the medium resolution and the use of a protein with little homology to build a model.

1F. We have responded to this comment above (see point 1B).

- the EM volume of the outer coat has been built from ~15K sub-volumes (~150 K sub-volumes for the inner coat). It might be useful to present the 3D classes obtained during the processing of the outer coat and discuss the local differences between the 3D classes. This may allow a better understanding of the consequences of local flexible zones on the flexibility of the whole outer coat. 1G. We regret the misunderstanding here. The various outer coat maps we present are not the result of classification, but of refining alignments of subtomograms extracted at different points in the tomogram. We performed classification only for the right-handed rods, as we show in Figure 4. We have added 'class 1' and 'class 2' to the Figure to distinguish the particular case where we did perform classification to the others, for which the picking procedure is explained in the Methods section.

Reviewer 1: The authors are correct, I apologize, it was not 3D classification rather than separate

groups of complexes.

- A map with local resolution for the inner coat should be presented to highlight the flexible regions in contact to the outer coat.

1H. We agree this is useful to highlight the inner coat flexible regions. We now include this in Supplementary Figure 2C.

Reviewer 1: this is useful.

Minor points

In the introduction, a figure presenting the COPII proteins and their role would be useful for non-expert COPII readers.

1I. We have added an introductory panel to Figure 7 which provides a summary view of the coat components. We believe adding this to Figure 7 provides a useful overview for non-expert readers, and at the same time it improves the clarity of our model presented in Figure 7B.

Reviewer 1: fine.

There is no figure sup 3 E. "Anti-Sec31 Western Blot of the 5-FOA derived SEC31 and sec31-NTD strains showing that the 835 expected size of Sec31 variants is present in the surviving cells".

1J. We apologise for the mistake. We have now removed this from the Figure legend (see also point 1A).

A representation of the EM envelope with the proposed fits for the inner and outer coat and membrane would be helpful to have an overview of the results (as depicted e.g. in figure 5 elife 2013).

1K. We already have figures with the models fitted into the cryo-EM maps. Due to the flexibility of each region with respect to the others, it is not possible to produce a map representing all coat layers simultaneously, and we feel the best way to represent them together is through a schematic representation, as in Figure 7.

Reviewer 1: Fine

Figure 7 is difficult to understand. A schematic 3D representation of the proteins and their interactions and with a lipid bilayer would be more useful.

1L. We feel adding a third dimension to the schematic of the protein and their interaction would make it even harder to understand. To improve clarity, we have added the membrane for reference and moved the labels next to the relevant proteins. We also added a panel (A) to provide an overview of the components (see point 1I), as well as summarising our findings on the function of outer cot assembly. We hope this improves the figure.

Reviewer 1: Fine

Reviewer #2 (Remarks to the Author):

The revision of the manuscript is reasonable.

However I would like the authors to write a few more words explaining Figure R2, i.e. the sharp increase in FSC value near Nyquist. They claim that this is a common occurrence in subtomogram averaging structures. It would be good to see the citations, because I do not understand it from an FSC theory perspective.

Nevertheless, I agree that their map probably is at the resolution they claim, so an explanation in words is probably enough at this stage.

Reviewer #3 (Remarks to the Author):

The clarifications and improvements to the revised manuscript have satisfactorily addressed my concerns and I am happy to recommend publication.

Please find below our response to further reviewer comments (our response in blue)

Reviewer #1 (Remarks to the Author):

The authors have answered to my questions and modified the manuscript in consequence. below are my new comments- starting by Reviewer 1:

Reviewer #1 (Remarks to the Author):

The report of J. Hutchings and colleagues is a structural study of the whole COPII complex polymerized on membranes by cryo-ET and subtomogram averaging.

The CopII machinery is involved in ER budding and vesicle formation for an anterograde traffic to the Golgi apparatus. Coatings are dynamic assemblies that assemble and dissociate at the ER. The COPII complex consists of 5 proteins, the GTPase Sar1 and the heterodimer Sec 23/Sec 24 form the internal coat while the heterotetramer Sec 13/Sec 31 forms the external coat. There are X-ray structures of Sec23, Sec24, Sar1 Sec31 active 683 peptide but no structure of the whole complex in the absence or presence of membrane. This limits our understanding of the mechanism of vesicle formation/assembly. The challenge of the structural analysis is that CopII is a multiprotein complex that forms variable supramolecular organization in the presence of membranes. The goal of the study is to describe the complete architecture of CopII to understand how CopII allows the formation of vesicles of different size and curvature.

This is a continuation of the work done by the same group that published in Elife 2013 and Nat Com 2018 with the same proteins and approaches by cryo-ET, subtomogram averaging and genetic of yeast. The in vitro system (yeast proteins, homologous expression, membrane system) are relevant. The cryo-ET and subtomogram approach is perfectly appropriate to answer the questions. The article is well written and the figures are well presented. The processing steps are very well described and performed with state of the art programs including a 3D CTF correction.

The article presents the molecular and supramolecular organization of the internal and external coats. The resolution is variable from 4.6 Å for the internal coat and 11-15 Å for the external coat. A protein organization is proposed by fitting the atomic models available in the EM envelope.

There is a very significant improvement in resolution on the outer coat from 40 Å to 11-15 Å (Elife 2013) and a slight improvement of the inner coat from 4.9 Å to 4.6 Å resolution (Nat Com 2018). The gain in resolution allows to confirm the arrangement of all proteins in each coat and to determine areas of interaction between the coats which are then discussed as being involved in the flexibility of the complex and the recognition of curvatures. In vitro biochemical and genetic experiments in yeast provide confirmation of the data derived from cryo-EM analysis.

The novelties are more specifically:

1) A better resolution of sec 13/Sec 31 which allows a fit of sec 13/31 structures. By comparing the atomic model and the EM envelope, it is proposed a localization for an acid loop that would stabilize interactions with neighbors. To validate this proposal, biochemical and genetic experiments are done with sec31 deltaNTD. However, the deletion does not prevent tubule formation, which is interpreted as the fact that interactions within the outer coat are not necessary to generate curvature on deformable membrane. Figure supp 3E of the in vivo experiments is missing in my document. It is also difficult to see if the order of the inner coat is preserved in the negatively stained NTD sec31 tubes. A cryo-EM image or better a 3D reconstruction would be more convincing. Given the resolution in the proposed Nter region, probably less than 12 Å, the assignment of the Nter domain of sec31 seems plausible but not certain. This point is important because if the Nter is positioned elsewhere, the proposition that the outer coating is not needed for generating curvatures is less valid.

1A. We regret a misunderstanding: the experiments with deltaNTD were carried out to determine the physiological importance of the vertex interaction and cage assembly for membrane deformation, and not to validate the stabilisation provided by the negatively charged loop. We did indeed find that membrane deformation can be achieved in the absence of the Sec31 N-terminal domain mediating outer coat vertex assembly, and that this is a condition that leads to productive budding in cells that have easily deformable ER membranes due to reduced cargo load.

We propose the membrane remodelling is driven by the inner coat, and in fact we can detect its ordered presence on tubules formed in vitro. The reviewer is correct in saying that the order is not immediately visible in the negatively stained images we present, so we have added an inset to Supplementary Figure 3D that shows the power spectrum of a typical coated tubule formed with Sec31 deltaNTD, where the inner coat lattice layer lines are indicated by arrows. This clearly shows inner coat order.

Reviewer 1: I agree with the explanations on the role of deltaNTD and the modification of the figure

No further change requested

As for the uncertainty of the assignment of the N terminus to the densities at the vertex, in Figure 1 we report the vertex structure, which was obtained by focussing the refinements on a small region, and indeed the reviewer is right that from this structure it is not clear that the assignment is unambiguous. However, at earlier stages of the processing, we obtain averages that encompass neighbouring vertices and interconnecting rods in a single low-

resolution map.

Given the fit of the rod-shaped X-ray model into these earlier maps, there is no ambiguity as to the identity of the domains at the vertex. We add here Figure R1 to illustrate this for the benefit of the reviewer. In addition, previous work utilising hydrogen-deuterium exchange has experimentally proved that the Sec31 NTD forms cage vertices (Noble et al, NSMB 2013).

Reviewer 1: I agree with the explanations. If possible to add the figure R1 in a supplementary figure, it could be useful for the reader.

We have added this as panel a to Supplementary Figure 3

Also, we apologise about panel S3E being missing, we had indeed removed it after internal revisions as we did not consider it essential, but forgot to remove the Figure legend. That panel contains a control experiment where we demonstrate that in emp24 depleted cells that grow in the presence of Sec31 deltaNTD, this is not due to Sec31 reverting to wild type. Unless required by the editor or reviewer, we would leave that panel out.

2) The organization of beta propellers in the vertex is different from that reported in the absence of a membrane for human sec13/31 complexes. The analysis of yeast COPII vertex on vesicle excludes a difference related to curvature and attributes this organization to the species difference. This experiment is convincing.

3) Two new densities of the outer coat are observed and attributed to domains that allow interactions between outer and inner coat: a) a new density extra density attached to the rod halfway between Sec13 and the dimerization interface. This density is attributed to the CTD domain of sec 13. A model of this domain is computed by homology with SRA1 and allows to propose a fit for the CTD domain in the EM envelop. However, the local resolution in this region of the EM does not allow a precise localization of the CTD domain. The lack of functional links between SAR1 and Sec31 questions the relevance of this modelisation. In vivo experiments confirmed the importance of this region for the stabilization of the external coat. b) additional and randomly distributed rods bridging between alpha-solenoid dimerization interfaces.

1B. We use Sra1 to build a homology model because we obtained significant similarity scores using HHpred searches. Moreover, we find that Sec31 CTD and Sar1 both belong to the same functional superfamily when searching the CATH database, and that the similarity between the respective subfamilies has an e-value which is widely considered beyond the threshold required for homology modelling. However, we agree with the reviewer that the functional and evolutionary link between these two proteins is unclear. To confirm the validity of our homology model we have used Robetta for ab initio structure prediction of the Sec31 CTD. When superimposed to the SRA1-derived model, the two structures are nearly identical, with a global RMSD of 2.3 Å, giving us confidence that the model we use is correct. We have modified the text as follows: "Since no atomic model for Sec31 CTD has been determined, we built a homology model to fit into the appendage density. Steroid Receptor RNA Activator protein (SRA1) 33 is a functionally unrelated protein that is found only in mammals, and its evolutionary links with Sec31 are unclear. Nevertheless, SRA1 and Sec31 CTD belong to the same evolutionary family and their similarity justifies the use of the SRA1 structure to build a homology model of the Sec31 CTD (see Methods). Rigid-body fitting the homology model in the appendage density shows consistency of size and features, although at this resolution we cannot determine the precise molecular interface (Supplementary Fig. 5A-C)."

Reviewer 1: The additional details in the Methods are important for the understanding of the strategy used for the modelisation of Sec31-NTD. Could you add if the homology model has been built using the mode ab initio, homology or deep learning TrRosetta of Robetta as the confidence value. Future works will definitively answer the validity of this prediction.

We have already specified above that this model was obtained using Robetta, ab initio option. We have not included this model in the manuscript, and we used it only as internal validation for the homology model in Supplementary Fig. 5

4) The internal coat is resolved to 4.6 Å resolution (4.7 Å written in the Table 3) which improves the current model at 4.9 Å, especially in the areas of interaction with the external coat. Thus known interaction regions are identified for the first time in the EM envelope: the region around the PPP sequence of sec31 which interacts with sec23 and a charged region of sec31 which interacts with sec23. In addition, a region of interaction between sec23 and sar1 is identified and validated by in vitro tubulation experiment.

1C. We have corrected table 3.

Overall the study is very well performed from the cryo-ET and image analysis point of view. This allows to obtain a more complete model of the COPII complex than the current models and highlights interaction zones. This is an additional example of the advances in subtomogram averaging of flexible multiproteic complexes associated with in vitro membranes. The resolution, which is very good for the inner coat, remains average for the outer coat, which leaves ambiguities on the organization of the proteins.

This is an important advance on the structural organization of CopII complex. However, the impact on the understanding of vesicle formation and stabilization seems less impressive likely because the resolution of the outer coat is not sufficient for an unambiguous assignment of all sub-domains of proteins required for building a

pseudo-atomic model. Biochemical and genetic experiments seem to confirm and complement the results of the cryo rather than providing original information. Thus, this report appears mostly as a structural study of a very good level and should be better considered in a journal such as Nature Structural and Cellular Biology. However, it could be considered for publication in Nature Communication after major modifications.

1D. We respectfully disagree with the reviewer in that the assignment of outer coat subunit is ambiguous. As we show above, we believe all domains that have been previously solved by X-ray crystallography can be assigned due to the peculiar rod-shape of the Sec13-31 complex. The one assignment that could not be made unambiguously from the fits (i.e. the Sec31 CTD) was validated by determination of the structure of a deletion mutant. We believe our biochemical and functional experiments add important insights, and do not just confirm the structural data. For example, the fact that either NTD or CTD deletion is lethal but rescued in emp24 delta background reveals that outer coat assembly functions to overcome membrane resistance rather than being the sole driver of membrane curvature. Also, our mutational analysis of the inner coat L-loop demonstrated how vesicle shape is dictated by the interplay of inner and outer coat assembly. These insights could not be obtained by the structure alone.

Reviewer 1: I agree with the explanations.

No further changes requested

Major modifications

- The role of the NTD needs to be better characterized. The density is not clearly defined in the EM envelope. The resolution should be improved if possible in this area, the sec31 deltaNTD tubes should be analysis and compare to WT and new genetic or biochemical approaches should strength the proposed role of the NTD sub-domain.

1E. We strongly believe that the resolution we obtained is enough to draw our conclusions on the assignment of the NTD in the EM map, we refer to point 1A above, including the newly added data on inner coat order in deltaNTD tubes. We're not sure what the reviewer would like to see regarding additional genetic and biochemical experiments. We already include EM analysis of GUV budding, microsome budding and cargo capture experiments, as well as viability assays in yeast. We also refer to our previous work using a N-his version of Sec31 which displays a weakened vertex interface, where the phenotype is more subtle, supporting cell growth and microsome budding but only in conditions where inner coat does not turn-over. We also show here that a functional vertex causes spherical vesicles to form when the inner coat lattice is weakened. All together we think these experiments clearly point to a role for the N-term as essential to reinforce the coat to overcome membrane resistance and drive towards spherical curvature in conditions where the inner coat can disassemble.

Reviewer 1: I agree with the explanations.

No further changes requested

- The CTD domain needs also to be better characterized. The density is defined in the EM envelope but the proposed model is questionable because of the medium resolution and the use of a protein with little homology to build a model.

1F. We have responded to this comment above (see point 1B).

- the EM volume of the outer coat has been built from ~15K sub-volumes (~150 K sub-volumes for the inner coat). It might be useful to present the 3D classes obtained during the processing of the outer coat and discuss the local differences between the 3D classes. This may allow a better understanding of the consequences of local flexible zones on the flexibility of the whole outer coat.

1G. We regret the misunderstanding here. The various outer coat maps we present are not the result of classification, but of refining alignments of subtomograms extracted at different points in the tomogram. We performed classification only for the right-handed rods, as we show in Figure 4. We have added 'class 1' and 'class 2' to the Figure to distinguish the particular case where we did perform classification to the others, for which the picking procedure is explained in the Methods section.

Reviewer 1: The authors are correct, I apologize, it was not 3D classification rather than separate groups of complexes.

No further changes requested

- A map with local resolution for the inner coat should be presented to highlight the flexibles regions in contact to the outer coat.

1H. We agree this is useful to highlight the inner coat flexible regions. We now include this in Supplementary Figure 2C.

Reviewer 1: this is useful.

No further changes requested

Minor points

In the introduction, a figure presenting the COPII proteins and their role would be useful for non-expert COPII readers.

1I. We have added an introductory panel to Figure 7 which provides a summary view of the coat components. We believe adding this to Figure 7 provides a useful overview for non-expert readers, and at the same time it improves the clarity of our model presented in Figure 7B.

Reviewer 1: fine.

No further changes requested

There is no figure sup 3 E. "Anti-Sec31 Western Blot of the 5-FOA derived SEC31 and sec31-NTD strains showing that the 835 expected size of Sec31 variants is present in the surviving cells".

1J. We apologise for the mistake. We have now removed this from the Figure legend (see also point 1A).

A representation of the EM envelope with the proposed fits for the inner and outer coat and membrane would be helpful to have an overview of the results (as depicted e.g. in figure 5 elife 2013).

1K. We already have figures with the models fitted into the cryo-EM maps. Due to the flexibility of each region with respect to the others, it is not possible to produce a map representing all coat layers simultaneously, and we feel the best way to represent them together is through a schematic representation, as in Figure 7.

Reviewer 1: Fine

No further changes requested

Figure 7 is difficult to understand. A schematic 3D representation of the proteins and their interactions and with a lipid bilayer would be more useful.

1L. We feel adding a third dimension to the schematic of the protein and their interaction would make it even harder to understand. To improve clarity, we have added the membrane for reference and moved the labels next to the relevant proteins. We also added a panel (A) to provide an overview of the components (see point 1I), as well as summarising our findings on the function of outer cot assembly. We hope this improves the figure.

Reviewer 1: Fine

No further changes requested

Reviewer #2 (Remarks to the Author):

The revision of the manuscript is reasonable.

However I would like the authors to write a few more words explaining Figure R2, i.e. the sharp increase in FSC value near Nyquist. They claim that this is a common occurrence in subtomogram averaging structures. It would be good to see the citations, because I do not understand it from an FSC theory perspective.

Nevertheless, I agree that their map probably is at the resolution they claim, so an explanation in words is probably enough at this stage.

We have added the following sentence to the methods:

"The FSC between refined half maps reveals an average resolution of 4.6 Å at 0.143 cutoff. We note a sharp increase in the FSC in correspondence to the Nyquist frequency. We were unable to find the source of that increase, but we are confident that the resolution reported is correct as shown by the local resolution map (Supplementary Fig. 2c)"

Reviewer #3 (Remarks to the Author):

The clarifications and improvements to the revised manuscript have satisfactorily addressed my concerns and I am happy to recommend publication.

No further changes requested